# Ultra-low power carbon nanotube/porphyrin synaptic arrays for persistent photoconductivity and neuromorphic computing

Jian Yao [1,2,6], Qinan Wang[2,3,6], Yong Zhang[2,6], Yu Teng[2], Jing Li [2], Pin Zhao[2], Chun Zhao [3] ✉, Ziyi Hu[2], Zongjie Shen[2], Liwei Liu[1,2], Dan Tian[4], Song Qiu [1,2], Zhongrui Wang [5], Lixing Kang [1,2] ✉ & Qingwen Li[1,2] ✉

Developing devices with a wide-temperature range persistent photoconductivity (PPC) and ultra-low power consumption remains a significant challenge for optical synaptic devices used in neuromorphic computing. By harnessing the PPC properties in materials, it can achieve optical storage and neuromorphic computing, surpassing the von Neuman architecture-based systems. However, previous research implemented PPC required additional gate voltages and low temperatures, which need additional energy consumption and PPC cannot be achieved across a wide temperature range. Here, we fabricated a simple heterojunctions using zinc(II)-meso-tetraphenyl porphyrin (ZnTPP) and single-walled carbon nanotubes (SWCNTs). By leveraging the strong binding energy at the heterojunction interface and the unique band structure, the heterojunction achieved PPC over an exceptionally wide temperature range (77 K-400 K). Remarkably, it demonstrated nonvolatile storage for up to $2\times10^4$ s, without additional gate voltage. The minimum energy consumption for each synaptic event is as low as 6.5 aJ. Furthermore, we successfully demonstrate the feasibility to manufacture a flexible wafer-scale array utilizing this heterojunction. We applied it to autonomous driving under extreme temperatures and achieved as a high impressive accuracy rate as 94.5%. This tunable and stable wide-temperature PPC capability holds promise for ultra-low-power neuromorphic computing.

In recent years, optoelectronic neuromorphic devices that mimic the functionality of the human retina system have attracted significant attention in the neuromorphic computing field[1-4]. These devices offer a higher information processing capability as well as a lower energy consumption. They aim to integrate visual perception, storage and computation[5-8], providing an ideal alternative to high-latency and high-power-consuming computing and storage based on the von Neumann architecture[9-12]. Nevertheless, there are still numerous

[1]School of Nano-Tech and Nano-Bionics, University of Science and Technology of China, Hefei 230026, China. [2]Advanced Materials Division, Suzhou Institute of Nano-Tech and Nano-Bionics, Chinese Academy of Sciences, Suzhou 215123, China. [3]School of Advanced Technology, Xi'an Jiaotong-Liverpool University, Suzhou 215123, China. [4]College of Materials Science and Engineering, Co-Innovation Center of Effiicient Processing and Utilization of Forest Resources, Nanjing Forestry University, Nanjing 210037, China. [5]Department of Electrical and Electronic Engineering, University of Hong Kong, Pokfulam Road, Hong Kong SAR 999077, China. [6]These authors contributed equally: Jian Yao, Qinan Wang, Yong Zhang. ✉e-mail: Chun.Zhao@xjtlu.edu.cn; lxkang2013@sinano.ac.cn; qwli2007@sinano.ac.cn

challenges in achieving this goal, with the most important one being the conversion and storage of light signals into nonvolatile electrical signals. At the material level, this phenomenon is known as a persistent photoconductivity (PPC). PPC plays a crucial role in simulating synaptic behavior under light stimulation in optoelectronic neuromorphic devices. It is also a key factor in achieving efficient and low energy consuming neuromorphic devices[10,13,14].

Heterojunctions comprising of various materials, including those based on 0D–1D materials[10], 2D materials[11,15–17], and 0D–2D materials[18,19], have been recently utilized to achieve PPC and mimic for neuromorphic computing. For each case, PPC is induced by either defect-induced carrier trapping[11,20,21], ion migration[10], or a low energy band barrier that suppresses carrier recombination at the interfaces between materials[13,22]. Typically, a gate voltage or low temperature conditions are required to achieve PPC, resulting in significant additional energy consumption. This does not provide any advantage compared to a range of 1–100 fJ per pulse for biological synapses. Additionally, most of the current research on these PPC phenomena has been conducted on hard substrates and individual devices, which is not conducive to subsequent flexible integration applications. Therefore, there is an urgent need to design and develop a high-quality heterojunction system. It should exhibit strong interactions and high energy band barriers to suppress carrier recombination, enabling zero gate voltage and stable PPC over a wide temperature range. Furthermore, this system should have the capability for large-area flexible integration and ultra-low power consumption.

Single-walled carbon nanotubes (SWCNTs) are considered ideal channel materials for next-generation electronic devices due to their high mobility[23]. Here, by utilizing zinc(II)-meso-tetraphenyl porphyrin (ZnTPP) as the primary light-absorbing layer and SWCNT the charge transport layer, and polyimide (PI) as the flexible substrate, we successfully achieved PPC in wafer-scale flexible device arrays. For a temperature as high as 400 K, ZnTPP can still firmly adhere to the surface of SWCNTs. Furthermore, due to their unique band structures, SWCNTs and ZnTPP create a barrier between their band, suppressing the recombination of photo-generated carriers. Therefore, the device can achieve PPC over a wide temperature range (77–400 K) without additional energy consumption. This characteristic was not present in previous devices, highlighting the robustness of the device under extreme temperature conditions. Additionally, the device can operate at ultra-low operating voltage ($10^{-7}$ V), with a minimum energy consumption of only 6.5 aJ per pulse. It is the lowest value known to us. We demonstrate the flexible wafer-scale integration capability of the heterojunction device and show the uniformity and stability of a $10 \times 10$ array. Finally, we apply it to the artificial retina system in autonomous driving, with a maximum prediction accuracy of 94.5%. Even in extremely low temperature (77 K) and high temperature (400 K) environments, the recognition accuracy can still be maintained above 90%. These prototype devices have almost unlimited tunability, paving the way for potential applications of novel multifunctional hybrid optoelectronic neural morphic devices.

## Results

### Design and fabrication of wafer-scale synaptic arrays

More than 80% of the information processed by humans daily is derived from the visual system. Figure 1a illustrates a schematic diagram of the human eye, it can convert light signals into neural signal. Further storage and processing take place in the human brain. Inspired by this, we designed a phototransistor using SWCNTs/ZnTPP as the active material to mimic the functions of the human eye and brain (Fig. 1b). In the phototransistor, ZnTPP serves as the optical absorption and electron transfer layer, while SWCNTs function as the primary conductive transport layer. The entire device is fabricated on a flexible polyimide (PI) substrate. Figure 1c illustrates an optical sensor array with 100 pixels (details of the fabrication can be referred to the

Experimental Section and Figs. S1–3 Supporting Information), showing its PPC characteristics at different temperatures, along with the corresponding neural network algorithm and a schematic diagram of autonomous driving. Figures 1d, e present optical photographs and details of the device array soldered onto the printed circuit board (PCB). Figure 1f shows the wafer-scale manufacturability of the flexible artificial synaptic device array, which demonstrates the large-scale integration capability of SWCNTs/ZnTPP phototransistors.

### Spectral and electrical characterizations on the SWCNTs/ZnTPP phototransistor

A pseudo-color SEM image of the device is shown in Fig. 2a, where the blue, green and yellow shaded areas indicate the source and drain electrodes, the gate and the channel region, respectively. The morphology of the SWCNT film is depicted in Fig. 2b, revealing the highly density and a remarkable uniformity, which is essential for a good device homogeneity and stability. Figure 2c showcases the optical absorption spectrum of ZnTPP in toluene, where a remarkable Soret band at around 422 nm and a Q band at near 550 nm can be found. The high absorbance of ZnTPP in the visible range indicates its potential application as a photosensitive material. The absorption spectra of the SWCNTs solution are depicted in Figure S4. It is evident from the analysis that the semiconductor purity exceeds 99.99%, making it ideal for application as a charge transport layer. The on-state current of the device decreased after spin-coating a layer of ZnTPP, along with a negative shift in the transfer curve as shown in Fig. 2d, indicating the injection of electrons from ZnTPP to SWCNTs. Compared to the SWCNTs/ZnTPP channel, the p-type SWCNT channel is more difficult to be turned on. The picoampere-level device leakage current demonstrates the robustness of the device. Figure 2e displays the photoluminescence of pure ZnTPP and that within the heterojunction of SWCNTs and ZnTPP. A pronounced quenching of the fluorescence is observed, indicating a strong built-in electric field between SWCNTs and ZnTPP, effectively inhibiting the recombination of hole-electron pairs. This phenomenon is in good agreement with the result of Raman spectroscopy in Fig. 2f, where the G mode peak of SWCNTs exhibits a shift of 2 cm$^{-1}$ after spin-coating ZnTPP. In the case of electron donation, the transfer of charge into the SWCNT π system induces a softening effect on the C – C bonds, leading to a redshift in the G peaks[24,25]. According to previous reports, SWCNTs readily undergo hole doping with water oxygen in air, exhibiting characteristics of p-type semiconductors[26]. The linewidth of full width at half-maximum (FWHM) can be controlled by the carrier concentration in SWCNTs[27–29]. Based on the transfer characteristic curve after spin-coating ZnTPP, it's evident that ZnTPP transfers some electrons to SWCNTs (i.e., weak n-type doping), thereby reducing the concentration of hole dopant and narrowing the linewidth. Moreover, the radial breathing mode (RBM) exhibited a blue shift of 1.9 cm$^{-1}$, which was caused by the π-π stacking interaction between SWCNTs and phthalocyanine aromatic rings[30,31], as shown in Figure S5.

### Charge storage capacity of the SWCNTs/ZnTPP phototransistor

Four different wavelengths of light are used to irradiate phototransistor, with a highest photocurrent being measured under 395 nm irradiation, as shown in Figure S6. For this reason, 395 nm is chosen as the excitation wavelength. The transfer characteristics of the phototransistor under different illumination conditions are depicted in Fig. 3a. With increasing light power, the transfer characteristics curve shows a noticeable positive shift. The fabrication of pure ZnTPP field effect transistor (FETs) was carried out using the spin-coating method. However, the device exhibited poor conductivity, as illustrated in Figure S7, further confirming the application of ZnTPP as a photosensitive layer. Figure 3b displays the output characteristics curve of the phototransistor, indicating a well-formed Ohmic contact between SWCNTs and Au electrodes. To verify the photosensitive effect of

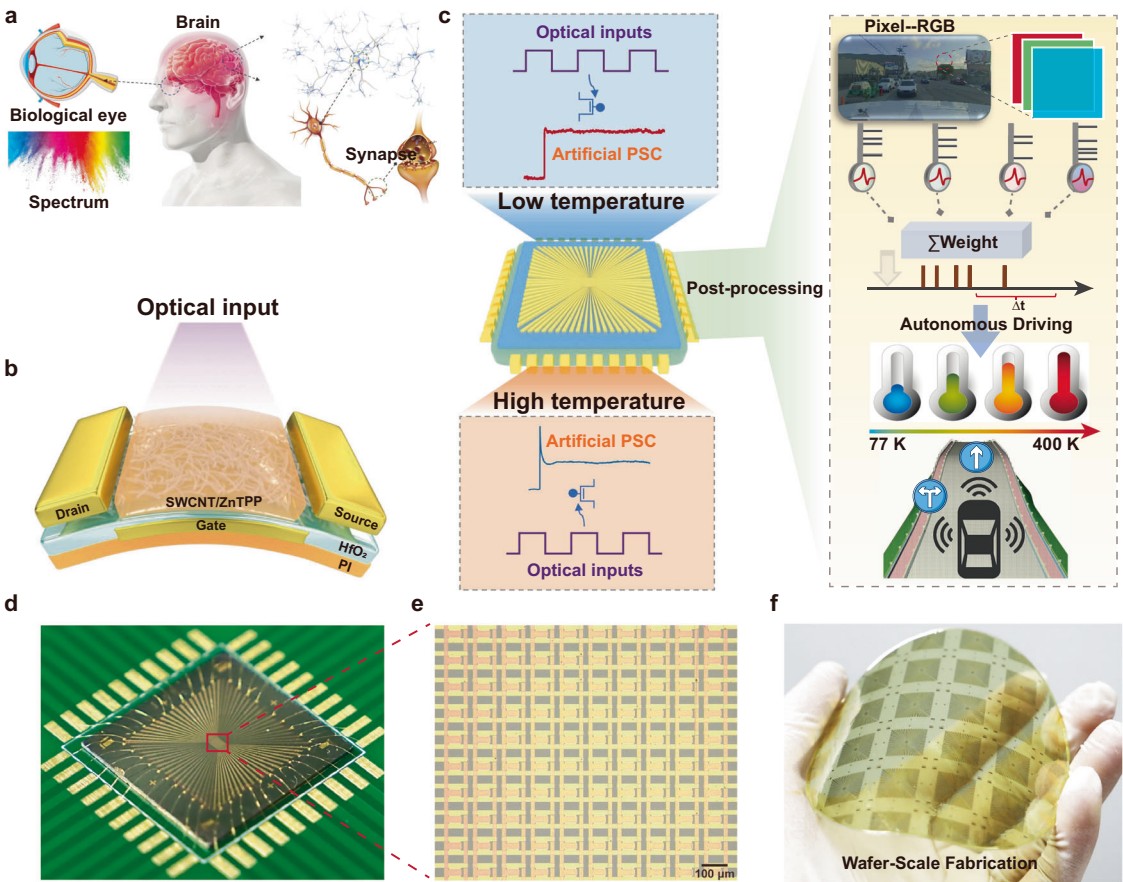

**Fig. 1 | Optical sensor arrays based on the SWCNTs/ZnTPP heterostructure for autonomous driving at different temperatures. a** Schematic diagram of a human visual perception system. **b** Schematic diagram of the SWCNTs and ZnTPP heterojunction device. **c** PPC characteristics of the artificial synaptic arrays under different temperatures and the corresponding schematic diagram of neural network algorithm and autonomous driving. **d** Wire bonding of flexible device arrays on a silicon substrate and printed circuit board connections. **e** Optical micrograph of a 10 × 10 device array (scale bar: 100 μm). **f** Optical photograph of a four-inch flexible wafer arrays peeled off from the silicon substrate.

ZnTPP, we tested the transfer characteristics of devices without spin-coated ZnTPP under the same light conditions, as shown in Figure S8. Pure SWCNTs devices did not exhibit a significant photoelectric effect due to a sufficiently high exciton binding energy[32]. It was difficult to separate the electron–hole pairs generated by light. However, when ZnTPP was spin-coated, the internal electric field at the heterojunction efficiently separated electron–hole pairs, resulting in a noticeable photocurrent. As shown in Fig. 3c, the SWCNTs/ZnTPP phototransistor was continuously irradiated with a laser of power of 1 mW/cm² for 30 s. Upon laser activation, the photocurrent sharply increased to a finite value, slowly decreasing after laser deactivation, and eventually stabilizing. The device maintained a reasonably high stability throughout the tested 2 × 10⁴ s period. Importantly, even after a 10 h power-off period, the device's current remained stable, indicating a remarkably strong nonvolatile charge storage capability. We also evaluated the influence of poly [9-(1-octylonoyl)−9H-carbazole-2,7-diyl](PCz) on the PPC characteristics of the device before and after washed. As shown in Figure S9, indicate that the PCz significantly affects the device's on-state current and PPC characteristics. Therefore, PCz cleaning before device fabrication is crucial. 450 nm, 532 nm and 620 nm lasers were also used as the light sources to determine the PPC characteristics of the phototransistor, as shown in Figure S10a–c which remained stable over the course of 2000 s, highlighting the universality of the device in the visible light range. Furthermore, in Figure S10d, the pulse-switching characteristics of optical potentiation and electrical depression were investigated in the SWCNTs/ZnTPP phototransistor under different light wavelength. The channel current of the transistor can be reversibly switched between high and low-current states under different light wavelength, demonstrating a high stability.

To gain deeper insight into the charge distribution of the device after illumination, surface potential tests were conducted using the Kelvin Probe Force Microscopy (KPFM). KPFM measurements were conducted in a nitrogen environment within a glovebox to exclude the influence of atmospheric water and oxygen. As shown in Fig. 3d, e, the surface potential of the device was approximately 120 mV under the dark condition. Subsequently, the phototransistor was subjected to laser irradiation with a duration of 30 s. After removing the light source, the surface potential was measured again, yielding a potential of 250 mV. This indicated that the surface of the SWCNTs/ZnTPP heterojunction carried negative charges. When the phototransistor was tested again after a 5 h interval, the surface potential dropped slightly to around 220 mV. This indicates that the charge distribution of heterojunction remained stable from the time being, demonstrating the device's exceptional nonvolatile charge storage capability.

Delving into the underlying mechanisms of this process, we conducted a comprehensive series of characterization tests on the phototransistor. As shown in Fig. 3f, to eliminate interference from water and oxygen in the air, vacuum and temperature tests were performed on the device. The results demonstrate that the device can achieve PPC under different temperatures, which is in contrast to the previously reported nonvolatile phototransistors that only operated at low temperatures[10,13]. Moreover, the low temperature test exhibited an increased stability since the temperature-induced thermal carrier disturbances was suppressed, as shown in Figure S11–13. At a temperature

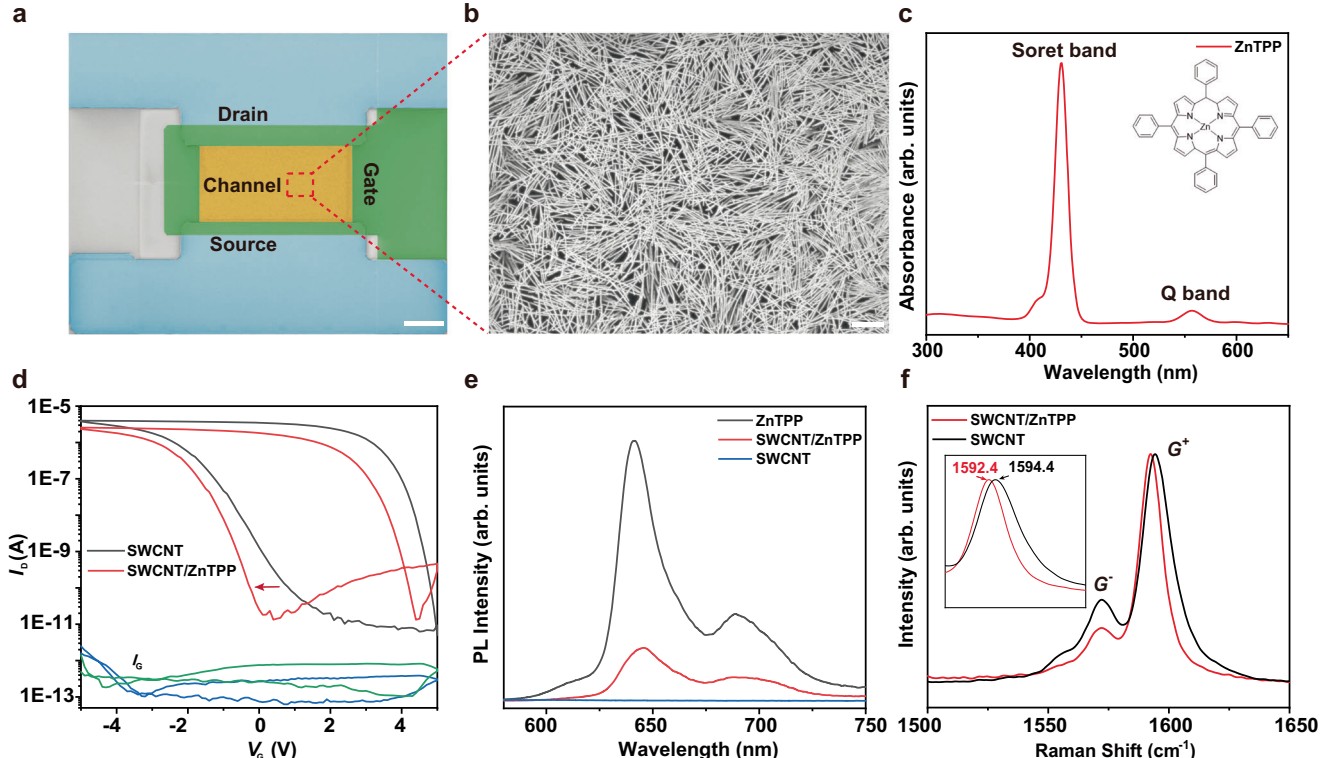

**Fig. 2 | Charge transfer between SWCNTs and ZnTPP. a** Pseudo-colored SEM image of a single device (scale bar: 10 μm). **b** SEM image of the SWCNT film in the device channel (scale bar: 1 μm). **c** UV-Vis absorption spectrum of ZnTPP. **d** Transfer curves of the device before and after spin-coating ZnTPP under a dark condition. (The blue line represents the leakage current before spin-coating of ZnTPP, and the green line represents the leakage current after spin-coating of ZnTPP). **e** Photoluminescence spectra of SWCNTs, ZnTPP, and SWCNTs/ZnTPP on a silicon oxide substrate. **f** Raman spectra of SWCNTs and SWCNTs/ZnTPP on a silicon oxide substrate, the inset shows an enlarged view of the SWCNTs G+ peak.

of 400 K, the storage capacity remained prominent. However, at 450 K, the device experiences a substantial loss in the storage capability. With the temperature reached 500 K, the storage capability disappeared completely. For a more precise investigation of the relationship between the storage capacity and temperature, we defined the following formula:

$$\eta = I_{light}/I_{dark} \qquad (1)$$

where $I_{light}$ represents the instantaneous current value immediately after the cessation of illumination, and $I_{dark}$ represents the stable current value after the cessation of illumination. We observed that as the temperature increased, $I_{light}/I_{dark}$ decreases consistently and disappeared at 500 K, as shown in Figure S14-15. This is attributed to the disturbance caused by thermal electrons as the temperature rises. Meanwhile, in the temperature range from 77 K to 350 K, the relationship between temperature and $\eta$ can be well fitted with a power-exponential function, which demonstrates a tunability of the SWCNTs/ZnTPP phototransistor. This result also rules out the mechanism of random local potential fluctuations caused by the intrinsic defects and charged impurity states.

The observed phenomenon was further investigated according to the theory of energy band distribution of SWCNTs and ZnTPP (Fig. 3g and Figure S16)[33–35]. Under the dark condition, SWCNTs exhibit a P-type conductivity characteristic, indicating the existence of numerous holes. When ZnTPP is coated on the SWCNTs channel, due to the higher Fermi level of ZnTPP compared to that of SWCNTs. Electrons in ZnTPP will transfer to the SWCNTs, which can be confirmed by the negative shift in the transfer curve of the SWCNTs in dark condition after spin-coating ZnTPP. It has led to the bending of the energy band of ZnTPP upwards and that of the SWCNTs downwards, forming a built-in electric field, which creates the energy barrier at the top of LUMO in ZnTPP near the interface. Upon laser illumination, ZnTPP rapidly generates a multitude of hole-electron pairs due to the photoelectric effect. The electron–hole pairs swiftly separate under the influence of an internal electric field, with holes injected into SWCNTs while electrons remain in ZnTPP. The process is similar to the previously reporte[36–38]. However, there is a difference that the barrier at the interface prevents the recombination of these electrons, which induced a persistent photocurrent in SWCNTs, as shown in Figure S17.

Because both SWCNTs and ZnTPP contain π electron clouds, there will be an overlap of π electron clouds when ZnTPP is spin-coated on the surface of SWCNTs. Through density functional theory (DFT) modeling, we calculated the binding energy ($E_b$) between ZnTPP and SWCNTs. The specific modeling method is elaborated in the experimental section. As depicted in Fig. 3h, we selected SWCNT with a chirality of (19,0) that has a diameter of approximately 1.5 nm, which is consistent with the average size of the semiconducting SWCNTs obtained by our method[39,40]. The results reveal that the adsorption distance of ZnTPP to SWCNTs is 3.34 Å, with an $E_b$ of 2.32 eV. This large binding energy indicates that the overlap of π electron clouds between SWCNTs and ZnTPP can result in a highly compact binding. As $k_BT$ is far less than $E_b$, ZnTPP can still adsorb firmly on the surface of SWCNTs even at a high temperature of 400 K, forming a grating effect. This also explains the reason why the device can achieve a storage of charge carriers at high temperature.

## Reconfigurable optoelectronic memory and neuromorphic synapse

Controlling the width and intensity of incident light pulses, we investigated the feasibility of optical programming and multistate data storage using the SWCNTs/ZnTPP phototransistor. Initially, light with a

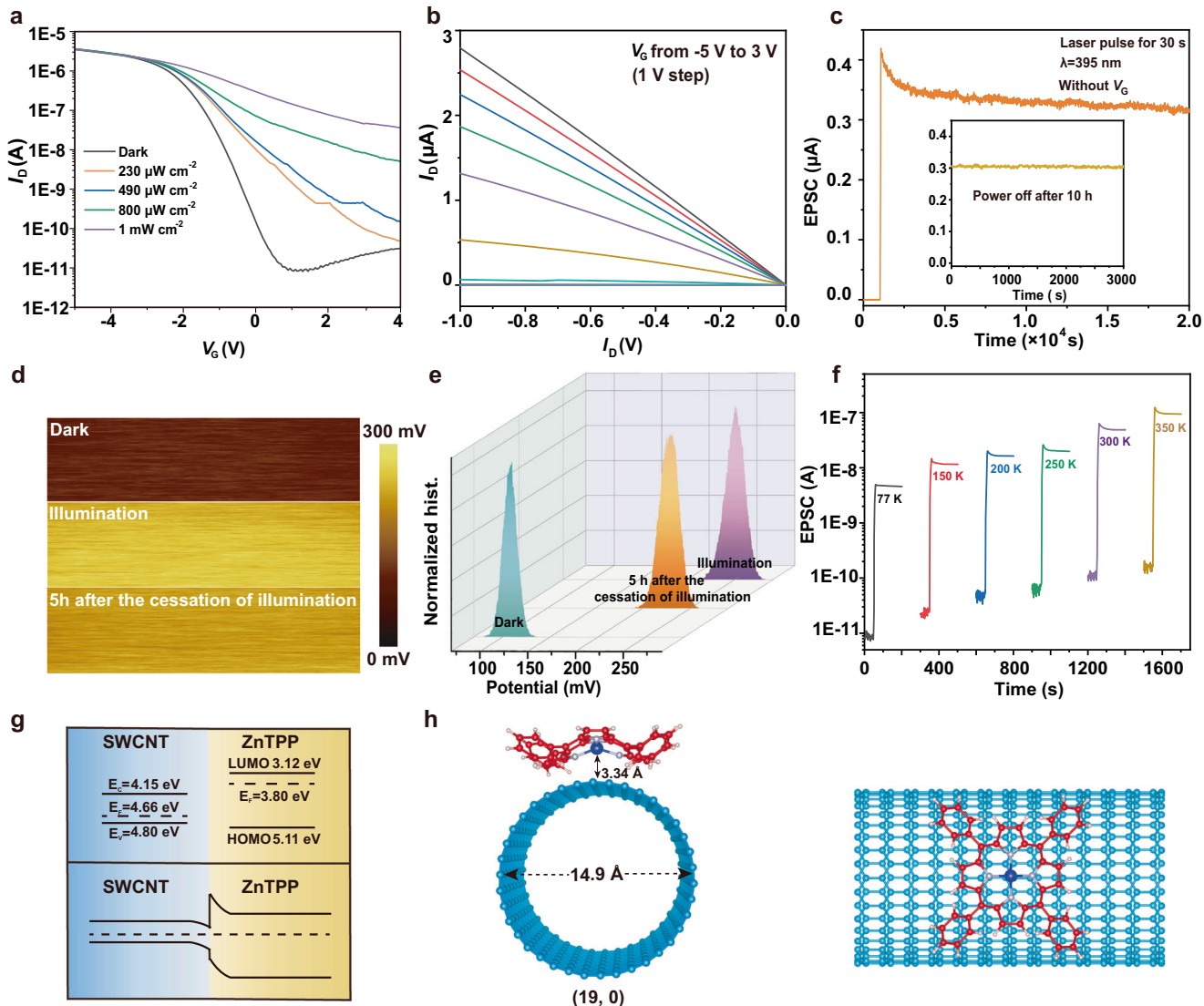

**Fig. 3 | Optical storage performance of the SWCNTs/ZnTPP phototransistor.** **a** Transfer curves of the device in the dark condition and after illumination with various light power ($V_{DS} = 1\,V$). **b** The output curve of the device in dark condition. **c** Optical pulse intrigues a PPC state. **d** Surface potential profile and **e** the potential distribution of the device recorded on the SWCNTs/ZnTPP heterostructure in the dark, upon illumination and 5 h after the cessation of the illumination, respectively. **f** PPC phenomena of the device induced by the same light pulse at varying temperatures. ($V_{DS} = 0.5\,V$) **g** Energy band structure of SWCNT and ZnTPP before contact and after contact under dark condition. **h** Schematic representation of ZnTPP molecules adsorbed on a SWCNT in the equilibrium geometry, front and top views of (19,0) SWCNTs/ZnTPP.

wavelength of 395 nm and a power of $1\,mW/cm^2$ with two pulse widths (0.1 s and 2 s) were used for writing, followed by erasure using a gate voltage of $-2\,V$, as shown in Fig. 4a. Notably, within the tested range of 10 cycles over 100 s, the device exhibited a stable optical writing and electrical erasure performance. As the pulse width of the light pulses increased, the device's photocurrent showed a relatively slow increase, demonstrating a high light tolerance. The repeatability of light response was measured separately at temperatures ranging from 77 K to 400 K, as shown in Figure S18, all exhibiting a good repeatability. As shown in Fig. 4b and Figure S19, the mechanism of this process is as follows[13,41]: at stage I, upon applying a laser to the device, the built-in electric field between SWCNTs and ZnTPP leads to the rapid spatial separation of photoexcited electrons and holes. This results in a swift increase in the carrier density $\Delta n$ in both SWCNTs and ZnTPP layers. Specifically, there is a notable increase in the hole concentration on SWCNTs, leading to an enhanced conductivity, while the electron concentration in ZnTPP also rises. At stage II, when the laser is turned off, the bending of the energy band in ZnTPP creates a barrier that

hinders the injection of electrons into SWCNTs; a certain number of electrons will also be trapped at the interface. However, after approaching this point, these electrons can continuously induce the generation of holes in SWCNTs through the photogating effect. At stage III, a negative $V_G$ pulse is applied, which can lead to the rapid depletion of photoexcited electrons in ZnTPP. At stage IV, as the negative $V_G$ pulse is turned off, $I_{DS}$ returns completely to its original state.

Continuously inducing current in SWCNTs through the photogating effect, thus serving as nonvolatile data storage as shown in Fig. 4c. Figure S20 shows a linear dependence of $I_{ph}$ on light pulse width and intensity, indicating a high efficiency of the optoelectronic synaptic transistor, underpinning its application potential as a precisely defined multistate memory device. In practice, by changing the pulse width and intensity of light pulses, more storage states can be obtained. Moreover, individual light pulses of different intensities can be used to directly turn the storage device to a specified state. Figure 4d indicates that the SWCNTs/ZnTPP phototransistor exhibits

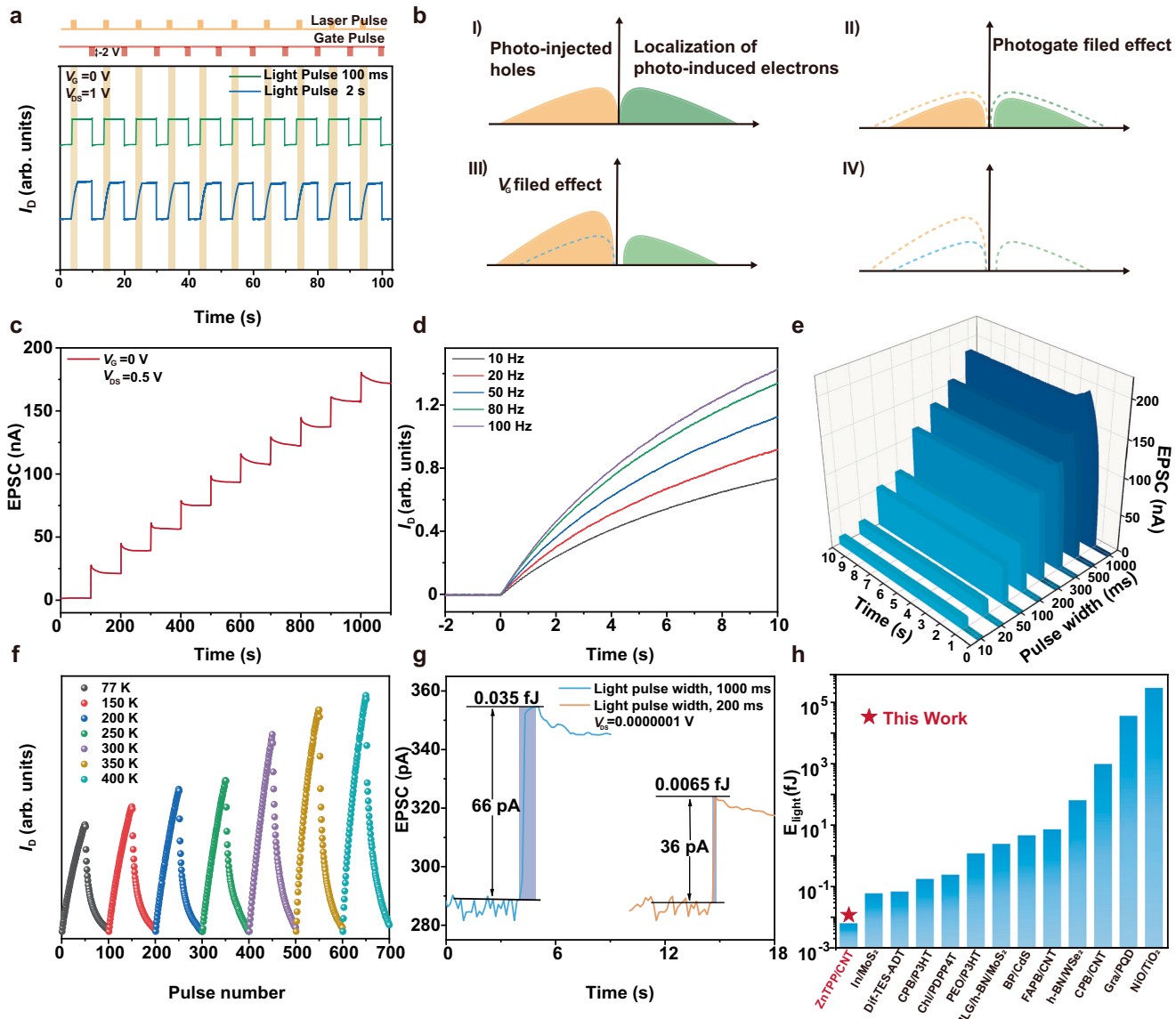

**Fig. 4 | Photoelectric programming and synaptic characteristics of SWCNTs/ ZnTPP phototransistors. a** Writing and erasing of a memory using optical pulses and gate voltage pulses, respectively. **b** Schematic representation of carrier distribution in the nonvolatile optical storage and erasure device with ZnTPP and SWCNTs heterostructure. **c** The nonvolatile multi-level conductance switching by applying 10 UV pulses. ($P = 0.2$ mW/cm²) **d** Photocurrent in ZnTPP/SWCNT heterojunction phototransistor as a function of time delivered at different frequencies.

**e** EPSC under different optical pulse width at a fixed light intensity ($V_{DS} = 0.5$ V, $V_G = 0$ V, $P = 0.8$ mW/cm², $\lambda = 395$ nm). **f** Light-controlled LTP ($P = 1$ mW/cm², duration of 1 s, spaced 1 s apart) and $V_G$ controlled LTD ($V_G = -1.5$ V, duration of 1 s, spaced 1 s apart) for 50 pulses under different temperature ($V_{DS} = 1$ V). **g** The lowest power consumption of the device is obtained as low as 6.5 aJ for a single pulse stimulus with pulse time of 200 ms and $V_{DS} = 0.0000001$ V. **h** The comparison of the device power consumptions with latest reported works.

temporal plasticity akin to biological synapses. As the pulse frequency increases from 10 Hz to 100 Hz, the synaptic weight transmitted after repeated pulses for 10 s almost doubled. The excitatory post-synaptic current (EPSC) of the SWCNTs/ZnTPP phototransistor can also be regulated by controlling the pulse width, as shown in Fig. 4e. Furthermore, in Fig. 4f, the pulse-switching characteristics of optical potentiation and electrical depression were investigated in the SWCNTs/ZnTPP phototransistor. The channel current of the transistor can be reversibly switched between high and low-current states at different temperatures, demonstrating a high stability and linearity.

Figure 4g demonstrates the power consumption data for device under the illumination of light pulses with different powers. Thanks to the excellent conductivity of SWCNTs and the outstanding photo-electric properties of ZnTPP, the device can detect effective EPSC under the illumination of light pulses at a $V_{DS}$ of $10^{-7}$ V. The power

consumption is calculated using the formula[11,42]:

$$E = V_{DS} \times I_{EPSC} \times t \qquad (2)$$

where $I$, $t$ represents the EPSC maximum current and $t$, respectively. At a pulse width of 1 s, the power consumption is 35 aJ. A prominent decrease in the pulse width to 200 ms still allows the device to generate effective EPSC, whilst the power consumption is as low as 6.5 aJ. Compared to those investigated in non-self-powered previous works[6,10,11,43–51], as shown in Fig. 4h, our photonic synaptic device exhibits the lowest power consumption and is significantly lower than the power consumption per synapse in human brain activity (1–100 fJ), indicating the device's enormous potential in developing ultra-low-power neuromorphic chips. Table S1 provides detailed results of device testing in this study and previous work, demonstrating the

superior capability of SWCNTs/ZnTPP in achieving PPC at wide temperature ranges and low power consumption compared to other material systems.

To validate the uniformity and stability of the device array, we subjected 1600 transistors in the array to transfer characteristic curve tests, as depicted in Figure S21. We recorded the corresponding on/off ratio, on-state current, and threshold voltage, revealing a satisfactorily high uniformity across our devices. 100 random sites were chosen for Raman measurements. The results are shown in Figure S22, where the RBM peak positions of the tested 100 points are approximately $171 \pm 0.6 \, cm^{-1}$, the $G^+$ peak are approximately $1594 \pm 0.4 \, cm^{-1}$. SEM provides images of more regions at the same time in Figure S23, showing a large-area uniformity of the 4-inch wafer. Figure S24 showcases device's resistance to the damage caused by bending. At a bent state, the maximum current under the illumination of light decreases proportionally. However, the device presented a stable storage characteristic, as demonstrated in Figure S24.

The transfer characteristic curves and storage properties of the devices exhibited no significant changes even after ten months of settlement, as evidenced in Figure S25. This underscores the durability and robustness of the flexible device array. We also extracted the long-term plasticity (LTP) and long-term depression (LTD) characteristic curves of the device under various temperatures, as presented in Figure S26–29.

## Simulation and modeling of autonomous driving at different temperature

In order to further examine the utility of our SWCNTs/ZnTPP photo-transistor arrays, we applied the array neural devices in the new energy unmanned vehicles[52,53]. To combine the ultra-low power characteristics of our devices and resistance to extreme environments, we chose SNN neural networks to simulate autonomous driving, thereby reducing overall power consumption and improving information fault tolerance. An autonomous vehicle refers to a vehicle that achieves an autonomous control and navigation of the vehicle based on sensors, controllers, deep learning, and actuators[54–56]. Research on the integration of autonomous driving technology and neural networks is a potential hotspot for intelligent applications. Importantly, the factors that affect the widespread distribution of autonomous vehicles are not only the computational efficiency, but also the external working environment (e.g., the temperature). Therefore, the impact of ambient temperature on the synaptic device directly affects the synaptic plasticity and stability[57–59]. Especially, the proposed transistors steadily demonstrate the synaptic plasticity at extremely low and high temperatures (from 77 K to 400 K). The six dynamic visual sensors carried by autonomous vehicles collect and process the road information to trigger the next action, as shown in Fig. 5a, b. Image information from six positions (back, front, front_left, front_right, left and right) is highly temporal related. Furthermore, spiking neural networks as the main algorithm structure for deep learning greatly reduce the energy consumption. The dynamic visual information is converted to multiple frames at a tunable rate. Therefore, input pulse data is obtained by sequentially encoding each red, green, blue (RGB) array from pixel values. The low energy consumption of the constructed spiking neural network is determined by a combination of the leaky integrate and fire (LIF) model, neural circuit policies (NCP), and spiking-timing-dependent plasticity (STDP) weight update rules[60]. The LIF model aims to simulate the working process of biological neuron models. When the membrane potential reaches the threshold $V_{th}$, the neurons trigger pulses, and the membrane potential will fall back to the resting value $V_{reset}$, which was shown in Fig. 5c. The refractory period in LIF effectively converts high-frequency information into low-frequency sparse information. Spiking information is transformed from image information through Poisson's equation at the temporal coding stage. NCP is sparse recurrent neural networks designed based on liquid time

constant (LTC) neuron and synaptic models. This strategy can significantly reduce the number of neurons and thus reduce energy consumption, as shown in Fig. 5d. The output pulse contains temporal information, which is consistent with the state of the human brain during operation. The output is regulated by STDP update rules[61,62]. Fig. 5e shows the pulse timing-dependent plasticity is a time asymmetric form of Hebb learning, guided by the tight temporal correlation between pre- and post-synaptic neuronal pulses. Like other forms of synaptic plasticity, STDP is widely believed to be the foundation for learning and information storage in the brain, as well as the development and refinement of neuronal circuits during the brain development. The time interval (Δt) in the STDP curve is the difference between the pre-synaptic pulse and the post-synaptic pulse[63,64]. In addition, the size of the Δt affects the weight adjustment scale in the SNN. The maximum scale of positive and negative feedback regulation is achieved when the absolute value of the difference reaches 10 ns. The synaptic plasticity of this neural device is measured at 6 different temperatures for the extremely low (77 K) and extremely high (400 K) temperature environment, whilst the corresponding hardware limitations ($G_{max}/G_{min}$, conductance levels and non-linearity) are embedded into the SNN. After 20 epochs of training, the prediction accuracy for the three directions (straight, turn left and right) of autonomous vehicles can reach 94.5%, as shown in Fig. 5f. The trend of LTD/LTP at different temperatures change the convergence time during the training phase. The convergence speed is the fastest for 300 K, and the overall convergence time is less than 150 s in Figure S30. During the continuous training and testing stages, the recognition accuracy can also be maintained at a high value in Figure S31. Meanwhile, the random fluctuations in the initial weight values affect the feedback process of the SNN. When the initial conductivity is set to change by 10%, the performance of the neural network is optimized, as shown in Figure S32. Furthermore, an appropriate capacity of parallel datasets can also improve the final recognition rate in Figure S33. The positive and negative connections between synapses are reflected by the weight values, and Figure S34 demonstrates in detail the weight arrays of some neurons after training. Faster convergence rate at room temperature is clearly observed, and the recognition rate remains ultimately over 90% in different environment (Fig. 5g), which exhibits an excellent tolerance to the extreme conditions. These experimental results indicate that such neural devices have the potential to adapt to various harsh conditions and be used for exploration in outer space.

Finally, we demonstrate the edge computing potential of our chip through a simple demonstration. As shown in the supplementary video, the car patrols the line and avoids obstacles. A patrol is an algorithmically planned route. All the algorithms are based on neural networks. Set up a camera above the car. The cameras collect information about the surrounding environment. The camera rotates once before each command is executed. The algorithm will give the next instruction according to the running result of the neural network. In the demonstration, the car avoided obstacles and followed the prescribed route, fully demonstrating the powerful potential of our chip.

## Discussion

In summary, our study demonstrates the scalable fabrication of SWCNTs and ZnTPP heterojunctions on flexible wafers for multifunctional neuromorphic brain-like computing. Notably, unlike previous studies, the interaction between SWCNTs and ZnTPP exhibits a large binding energy and unique photoelectric performance. Electrons generated by photoexcitation can be stored in ZnTPP, resulting in a photogating effect. The device can achieve persistent photoconductivity between 77 K to 400 K. Leveraging the strong light absorption coefficient of ZnTPP and the high electrical conductivity of SWCNTs, the device can be operated at a $V_{DS}$ of only $10^{-7}$ V, which corresponds to an extremely low power consumption of 6.5 aJ per pulse. Moreover, our simulations successfully emulate synaptic

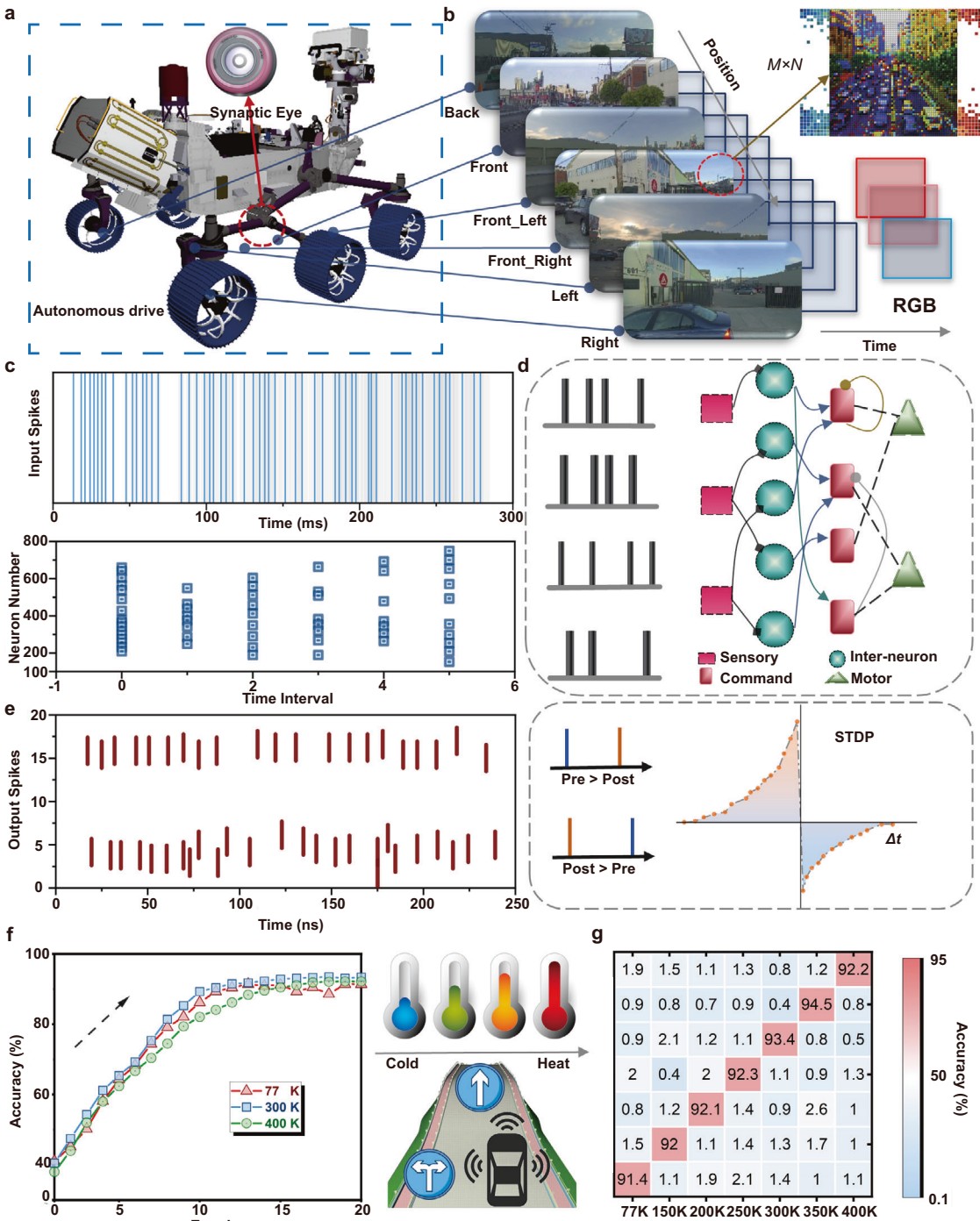

**Fig. 5 | Simulation and modeling of synaptic arrays for autonomous driving.**
**a** The autonomous vehicle is equipped with six visual sensors for autonomous driving. **b** Data-set of image information is extracted by RGB values of each pixel point. **c** The process of temporal encoding and LIF filtering of high-frequency pulse signals. **d** Basic framework of neural circuit policies for spiking neural network.

**e** Feedback propagation in SNN based on STDP trend as weight update rule.
**f** Change in recognition accuracy with increasing training epoch under extreme temperatures (77 K and 400 K). **g** Confusion matrix at various temperatures for dynamic recognition.

behaviors (EPSC, LTP, LTD) on a 10 × 10 flexible array. In unmanned driving systems, the prediction accuracy of the flexible array reaches 94.5%, with all the recognition accuracy remaining above 90% across a wide temperature range (77 K to 400 K), demonstrating extraordinary robustness. The unique wide temperature tolerance and wafer-scale manufacturing capability of the SWCNTs and ZnTPP heterojunction phototransistors pave the way for large-scale integration and practical applications in neuromorphic devices.

## Method

### Preparation of highly pure SWCNTs dispersed in toluene

In all of 10 mg of poly [9-(1-octylonoyl)−9H-carbazole-2,7-diyl] (Mn = 45,000 Da) and 20 mg of pristine arc-discharge carbon nanotubes were added to 20 mL of toluene. Tip sonication with a power of 50 W was applied for 1 h. The resulting solution was subjected to centrifugation at 40,000 g for 1 h. Finally, the supernatant from the centrifuged solution was collected, resulting in a high-purity dispersion of

SWCNTs. All other materials were obtained from Sigma Aldrich without further purification.

## Device Fabrication

The polyimide (PI) film used in this study, model ZKPI-302, was purchased from POME Technologies. It was spin-coated onto conductive silicon wafers at 3000 rpm for 30 s. The cured PI substrate was transferred to a glovebox and baked at 300 °C for 1 h to ensure complete solidification. Standard photolithography techniques were employed to fabricate gate electrodes (Ti/Au, 10/50 nm) on the cured PI substrate. An atomic layer deposition (ALD) method was utilized to grow a 70 nm thick layer of $HfO_2$ which was employed as the gate dielectric layer at 250 °C. Subsequently, the dielectric layer is exposed and etched by the lithography and ion beam etching techniques to expose the grid to prepare the connections. 20 ml of the SWCNT solution was diluted to 100 ml in prior to thin film network deposition based on an immersing method. With 2 h of deposition, a high-density and random network of SWCNTs can be obtained. The polymer on the surface of SWCNTs was then removed by the tetrahydrofuran cleaning. The clean SWCNTs network film was then patterned using the standard photolithography techniques, with excessive SWCNTs removed by $O_2$ plasma etching. Source and drain electrodes (Ti/Au, 10/50 nm) were fabricated based on electron beam evaporation. Finally, a solution of ZnTPP was spin-coated onto the surface of the SWCNTs transistor arrays. The detailed steps can be found in the supporting information.

## Characterization and Measurements

Optical absorption spectra were measured on a UV-Vis-NIR spectrophotometer (Lambda 750). Scanning electron microscope (SEM) images were characterized by a Gemini300 field-emission instrument. The atomic force microscope (AFM) images were characterized by a Dimension ICON with a tapping mode. Kelvin Probe Force Microscopy (KPFM) measurements were carried out on a Cypher S (Asylum Research, Oxford Instruments) in combination with a HF2LI lock-in amplifier (Zurich Instruments) using Cr/Au-coated conducting probes (NSC14, Mikromasch). Raman characterization was measured by a Lab RAM HR-800 Raman spectrometer with a laser excitation wavelength of 532 nm. The electrical performance of the device was evaluated using a Keithley 4200 semiconductor parameter analyzer, which equipped with a vacuum chamber, laser source and temperature control system.

## DFT calculations

We used the DFT as implemented in the Vienna Ab initio simulation package (VASP) in all calculations. The exchange-correlation potential is described by using the generalized gradient approximation of Perdew–Burke–Ernzerhof (GGA-PBE). The projector augmented-wave (PAW) method is employed to treat interactions between ion cores and valence electrons. The plane-wave cutoff energy was fixed to 500 eV. Given structural models were relaxed until the Hellmann–Feynman forces smaller than −0.02 eV/Å and the change in energy smaller than $10^{-5}$ eV was attained. During the relaxation, the Brillouin zone was represented by a Γ centered k-point grid of $10 \times 10 \times 10$. Grimme's DFT-D3 methodology was used to describe the dispersion interactions among all the atoms in adsorption models.

## Spiking neural network

The spiking neural network is connected by pulse neurons, whose inputs and outputs are the series pulses. The pulse neurons have the leaky integrate and fired (LIF) model. When the electromotive force increases faster than the decay rate (such as frequent pulse inputs), the electromotive force inside the neuron will increase until it reaches a certain firing threshold, and the neuron will emit a pulse. Meanwhile, the refractory period of LIF can convert high-frequency information into low-frequency sparse information. The input signal is to convert dynamic visual information into pulses through temporary coding. Furthermore, during the weight update process, the fluctuation of weight values is based on the spike timing-dependent plasticity (STDP) rule. Mainly relying on the difference between pre-synaptic pulses and post-synaptic pulses. The neural networks and strategies used are based on well-established SnnTorch and NCP[64,65]. Additionally, we preprocess the video data to match the input type required by the networks.

## Date availability

The data that support the plots within this paper and other finding of this study are available from the corresponding author upon request. Source data are provided with this paper.

## Code availability

All code used in simulations supporting this article is available from the corresponding authors upon request.

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

## Acknowledgements

J.Y., Q.W. and Y.Z. contributed equally to this work. This work was supported by the National Natural Science Foundation of China (52372055, 52272136, 22175094, 21971113), the Key Research and Development Program of Jiangsu Province (BK20232009), the Natural Science Foundation of Jiangsu province (BK20221402), the Basic Research Development Program of SuZhou (SJC2023004), Jiangsu Province Youth Fund (BK20220289), the China Postdoctoral Science Foundation (Grant No. 2022M720104). The authors are grateful for the technical support for Nano Fabrication Facility and NANO-X from Suzhou Institute of Nano-Tech and Nano-Bionics, Chinese Academy of Sciences (SINANO).

## Author contributions

Q.L., L.K., and C.Z., conceived the idea and supervised the experiment. J.Y. designed the devices, performed the device fabrication and properties characterization with the assistance of Y.T. and Z.S. Q.W. and Z.W conducted simulations for autonomous driving. S.Q. and L.L. provided the SWCNTs. J.L. and D.T. provided the mechanism analysis. P.Z. conducted the KPFM characterization. Z.H performed the Raman and fluorescence characterizations. J.Y., Q.W., and Y.Z. prepared the figures and wrote the manuscript. All authors discussed the results and commented on the manuscript.

## Competing interests

The authors declare no competing interests.
