## [Peer Review File · Nature Communications]

Ultra-Low Power Carbon Nanotube/Porphyrin Synaptic Arrays for Persistent Photoconductivity and Neuromorphic ComputingREVIEWER COMMENTS

Reviewer #1 (Remarks to the Author):

The authors systematically presented the work. However, the concept of device structure is not novel. Similar device structures with the photo sensing material (Zn-TPP) works have already been reported [10.1021/nl0630485] and [10.1021/nl061231s]. The following second reference is missing. Additionally, the SiNWs have also proven to be thermally stable. Moreover, optical switching of Zn-TPP coated SiNWs and SWCNTs has already been reported back in 2007 and 2006. The authors have improvised by showcasing the flexibility and large-area processing of the devices with high-temperature attributes. The authors need to comment on this.

Nevertheless, the authors tried to show some interesting aspects and practical applicability of ultra-low-power neuromorphic computing. Based on the responses from the authors to my comments, the editor can decide the suitability of publishing it.

Comments:

1. The gate leakage current of both the transfer characteristics is needed in Figure 2d.
2. Figure 2f. is missing in the main content, cite wherever necessary.
3. Elaborate more on the reason for the blue shift and the decrease in the FWHM (G+) for the heterojunction case.
4. It is recommended to show the transfer characteristics of bare Zn-TPP.
5. Revise the energy band diagram with the fermi level showing on either side.
6. Schematically illustrate the energy band bending for different stages as discussed in the manuscript.
7. References need on lines 31-33

Reviewer #2 (Remarks to the Author):

In this manuscript, the authors reported a hybrid ZnTPP/SWCNT heterojunction with strong binding energy. This junction exhibits wide-temperature nonvolatile optical memory and neuromorphic computing with ultra-low power consumption. However, this distinctive features are not fully supported by its related characterizations and data, and the method's uniqueness warrants further investigation. Several questions should be resolved as following.

1. The authors demonstrated that the PPC behavior was attributed to charge trapping between semiconductive SWCNT and ZnTPP. Generally, SWCNTs contain semiconductive and conductive types. What is the purity of semiconductive SWCNTs in this manuscript?
2. Due to the advantage of wafer-scale manufacturing process, the reproductivity of each cells is very important, please demonstrate the reproductivity.
3. In the figure 3f, the photoresponse properties are highly dependent on the working temperature, thus the title with wide-temperature optical memory is seem to be inaccurate. Meanwhile, the recyclability of temperature-dependent photoresponse should be measured.
4. The authors employ SEM image to evaluate the uniform of film, and XRD or Raman spectra from different areas should be also provided.
5. To highlight the nonvolatile feature of the heterojunction device, the authors are suggested to prolong the retention time.
6. The mechanism of nonvolatile behavior should be explained in more details.
7. In the figure 4h, the comparison of the device power consumptions with latest reported literatures should summarize the results of self-powered works.
8. The optical absorption spectra of ZnTPP is at 422 nm and 550 nm, while the authors choose 395 nm as the pulse light, why?
9. In the figure 5b, the authors demonstrate that the data is obtained by encoding RGB array from pixel values, thus the EPSC measurement at different wavelength of pulse light should be provided.

Response Letter to Reviewers' Comments

We sincerely appreciate the reviewers for carefully reading our manuscript '*Ultra-Low Power Carbon Nanotube/Porphyrin Synaptic Arrays for Wide-Temperature Optical Memory and Neuromorphic Computing*' (NCOMMS-24-04786) and providing constructive comments. According to the reviewers' comments, we have carefully revised our manuscript and provided more detailed data to improve the quality and readability of the manuscript. We rearranged the manuscript and supplementary information by inclusion of new data with elaborate discussions and abundant experiment data. With the help of the reviewers, we believe that the submitted version has been significantly improved.

Following the reviewer's valuable suggestion, we have adjusted the title. The new title is: '*Ultra-Low Power Carbon Nanotube/Porphyrin Synaptic Arrays for Optical Memory and Neuromorphic Computing*'. Below are the point-by-point responses to each comment.

The corresponding revisions concerning comments have been provided and highlighted in red in the revised manuscript and supplementary information. The corresponding responses of the reviewers are marked by blue words.

Reviewer #1

General comment:

The authors systematically presented the work. However, the concept of device structure is not novel. Similar device structures with the photo sensing material (Zn-TPP) works have already been reported [10.1021/nl0630485] and [10.1021/nl061231s]. The following second reference is missing. Additionally, the SiNWs have also proven to be thermally stable. Moreover, optical switching of Zn-TPP coated SiNWs and SWCNTs has already been reported back in 2007 and 2006. The authors have improvised by showcasing the flexibility and large-area processing of the devices with high-temperature attributes. The authors need to comment on this. Nevertheless, the authors tried to show some interesting aspects and practical applicability of ultra-low-power

neuromorphic computing. Based on the responses from the authors to my comments, the editor can decide the suitability of publishing it.

Response:

We sincerely appreciate the reviewer for dedicating time and effort to thoroughly review our paper and consult relevant literature. As noted by the reviewer, the concept of device structure elucidated in this work may lack the novelty and the device structure is straightforward. However, we contend that its simplicity can garner increased interest due to the convenience to reproduce. Furthermore, in contrast to previous works, we have advanced our research into the emerging field of neuromorphic computing, which marks a significant breakthrough. In the post-Moore era, there is an increased demand for low-power and integrated architectures for both sensing and computing (Mennel, L. *et al. Nature* 579, 62, 2020; Wu, G. *et al. Nat. Mater.* 22, 1499, 2023). We are confident that our work will certainly pique the interest of a broad readership and pave the way for large-scale flexible integration and practical applications of the neuromorphic devices. We provide further clarification as outlined below.

First, we are sorry that we have missed this reference, and the citation has been added in the revised manuscript. We compare our work with previous studies, as shown in **Table R1**. Previous studies have indeed utilized SWCNTs as the charge transport layer of transistors, and porphyrin hybrids have also been employed as the photosensitive layer. However, they were limited by the semiconducting type purity of the carbon nanotubes they used, as indicated by the relatively low transistor on/off ratio (10^2), indicating the existence of metallic tubes in the SWCNTs used. Similarly, due to a difference in the molecular structure of porphyrins, these studies did not show the same persistent photocurrent (PPC) as in our work, nor did they achieve subsequent flexible wafer-scale integration and neuromorphic computing applications. Additionally, while Si NWs have been demonstrated to exhibit a sufficiently high thermal stability, it is important to clarify that the focus of our work lies in the ability to achieve the optical storage over a wide temperature range (77 K-400 K), while also

enabling the ultralow power neuromorphic computing, a capability that haven't been demonstrated by SiNWs/ZnTPP heterojunctions.

In this work, we fabricated a simple heterojunction using ZnTPP and SWCNTs. The heterojunction achieved optical memory over an exceptionally wide temperature range (77 K-400 K). This has not been reported in previous papers. Remarkably, it demonstrated a nonvolatile storage over the course of up to 2×10^4 s, without applying an additional gate voltage. The minimum energy consumption for each synaptic event is as low as 6.5 aJ. Furthermore, we successfully demonstrate the feasibility to manufacture a flexible wafer-scale array utilizing this heterojunction. We applied it to autonomous driving in extreme temperatures and achieved an impressive accuracy rate of 94.5%. We also compare our work with relevant studies¹⁻⁹, as shown in **Table R2**. Our devices offer significant advantages in flexible wafer-scale integration, optical memory, and demonstrate an excellent performance in terms of storage duration and power consumption.

Table R1. Comparison of previous paper and this work.

Comparison	Previous Paper 1 [10.1021/nl0630485]	Previous Paper 2 [10.1021/nl061231s]	This Work
Optical memory	No	No	Yes
Wafer-scale flexible integration	No	No	Yes
Neuromorphic computing	No	No	Yes

Table R2. Comparison of devices with persistent photoconductivity for latest Literature Studies. (RT: Room temperature)

Active materials	Substrate	Temperature (K)	Time (s)	Energy Per Spike (fJ)	Scale	Reference
SWCNT/ FAPbBr ₃	Si/SiO ₂	~50-RT	~5×10 ³	7.4	Single	Ref.1
BP/CdS	Si/SiO ₂	RT	> 5×10 ³	4.8	Single	Ref.2
MoS ₂ /PbS	Si/SiO ₂	<200	> 1×10 ⁴	138000	Single	Ref.3
IGZO/PVK	PET/Al ₂ O ₃	RT	> 1×10 ⁴	Not mentioned	~2 cm×2 cm	Ref.4
MoS ₂ /PO _x	Si/SiO ₂	80-300	> 1×10 ⁴	Not mentioned	Single	Ref.5
Pentacene/ CsPbBr ₃	Si/SiO ₂	RT	~3×10 ³	~140	Single	Ref.6
VO ₂	Si/SiO ₂	RT	~4×10 ³	Not mentioned	2-inch	Ref.7
PDPP4T/ chlorophyll	Si/SiO ₂	RT	~4×10 ²	0.25	Single	Ref.8
Graphene/ PQD	Si/SiO ₂	RT	~3×10 ³	37000	Single	Ref.9
SWCNT/ ZnTPP	PI/HfO₂	77-400	>2×10⁴	0.065	4-inch	This work

Here are the key improvements compared to previous paper:

(1) In terms of materials, after calculating the purity of semiconducting carbon nanotubes used in this work, it was found to be as high as 99.99%. This contrasts starkly with previous studies (10.1021/nl061231s), where the device's on/off ratio was only about 10², indicating the presence of metallic tubes in the film, resulting in a lower device on/off ratio. In contrast, data from 1,600 devices in this study showed an on/off ratio of approximately 10⁵-10⁶. High-purity semiconducting carbon nanotube serve as the foundation for achieving persistent photoconductivity in SWCNTs/ZnTPP heterojunctions. The absence of persistent photoconductivity in previous investigations can now be attributed to the lower purity levels of the carbon nanotube films employed, highlighting the importance of stringent material characterization and selection in

research endeavors.

(2) In terms of device integration, in previous studies, most research focused on investigating the performance of individual devices, as shown in **Table S2**, which constrained the scalability and large-scale integration capabilities of related applications. Although some studies reported a wafer-scale integration, they typically utilized rigid substrates, which limits the flexibility and versatility of device integration. To the best of our knowledge, we are the first to report on a flexible wafer-scale integration based on SWCNTs/ZnTPP heterojunctions and have successfully demonstrated the persistent photoconductivity effect using a simple heterojunction, which has been effectively applied in neuromorphic computing.

(3) In terms of application, neuromorphic devices have great potential in many artificial intelligence domains, especially addressing the energy bottleneck of current von Neuman architectures. Achieving real-time autonomous driving requires breaking through the energy barrier of chips. Leveraging the biomimetic advantages of neuromorphic devices, they can think like the human brain and operate with extremely low power consumption. Neuromorphic chips simulate the neural and synaptic structures of the human brain, offering high energy efficiency and enabling large-scale parallel computing at low power consumption, crucial for processing vast sensor data and complex algorithms in autonomous driving systems. The parallel computing architecture of neuromorphic chips allows for highly concurrent processing, enabling real-time handling of multiple sensor data inputs, rapid decision-making, and real-time adjustments. Neuromorphic chips possess certain adaptive learning capabilities, improving system performance through continuous training and feedback, adapting to various driving scenarios and road conditions. Their structure allows for brain-like plasticity, providing self-learning and adaptability to rapidly adjust to different driving environments. Neuromorphic chips also exhibit resistance to interference, effectively handling noise and disturbances in sensor data, enhancing the stability and reliability of autonomous driving systems.

Furthermore, we combine spiking neural networks with network channel pruning (NCP). NCP is a technique aimed at significantly reducing the number of neurons while

maintaining network performance. By employing this strategy, we can construct more compact neural network structures, further reducing overall energy consumption. When NCP is combined with spiking neural networks, it can further reduce energy consumption, particularly suitable for energy-sensitive applications such as autonomous driving. NCP can reduce network complexity by removing redundant neurons and connections and streamlining network hierarchy, reducing computational and communication energy consumption. NCP can utilize fine-grained weight representation methods, such as low-precision weights or compression techniques, reducing parameter storage and transmission costs in neural networks. NCP can leverage heterogeneous computing architectures, assigning different types of computing tasks to suitable processing units to improve efficiency and reduce overall energy consumption. Autonomous driving systems typically operate on energy-limited embedded devices, thus imposing strict requirements on energy consumption. Combining NCP with spiking neural networks can effectively reduce system energy consumption, extending system battery life or reducing energy consumption. Autonomous driving systems require processing complex perception data, such as images, radar, and LiDAR data. Combining NCP with spiking neural networks can provide efficient perception and decision-making capabilities, helping autonomous driving systems cope with various complex driving scenarios.

In summary, inspired by biological systems, we present a multifunctional heterojunction between ZnTPP and SWCNTs. Compared to other studies, it exhibits several notable advantages as followed:

1) Firstly, **it exhibits stable storage over an extremely wide operating temperature range (77 K-400 K) without gate bias and achieves multi-state non-volatile storage.** These results arise from the strong binding energy and unique band structure built between SWCNTs and ZnTPP.

2) Secondly, due to the strong light absorption coefficient of ZnTPP and high electrical conductivity of SWCNT, **the heterojunction can operate efficiently at ultralow voltage (10^{-7} V), with a minimum energy consumption as low as 6.5 aJ per synaptic event, which is the lowest reported in non-self-powered previous**

works.

3) Finally, for the first time, we successfully fabricated a 4-inch-scale flexible wafer array based on SWCNTs/ZnTPP. **In simulated autonomous driving, combined with low-power neural networks, the recognition accuracy reached 94.5%. While operating under various extreme temperature conditions, recognition accuracy of over 90% can be achieved.**

Overall, we greatly appreciate the reviewer's insightful comments and welcome the opportunity to address their concerns while offering further context for our work.

References:

1. Hao, J. *et al.* Low-energy room-temperature optical switching in mixed-dimensionality nanoscale perovskite heterojunctions. *Sci. Adv.* **7**, eabf1959 (2021).
2. Zhu, C. *et al.* Optical synaptic devices with ultra-low power consumption for neuromorphic computing. *Light. Sci. Appl.* **11**, 337 (2022).
3. Wang, Q. *et al.* Nonvolatile infrared memory in MoS₂/PbS van der Waals heterostructures. *Sci. Adv.* **4**, eaap7916 (2018).
4. Wei, S. *et al.* Flexible Quasi-2D Perovskite/IGZO Phototransistors for Ultrasensitive and Broadband Photodetection. *Adv. Mater.* **32**, 1907527 (2020).
5. Liu, C. *et al.* Realizing the Switching of Optoelectronic Memory and Ultrafast Detector in Functionalized-Black Phosphorus/MoS₂ Heterojunction. *Laser & Photonics Rev* **17**, 2200486 (2023).
6. Wang, Y. *et al.* Photonic Synapses Based on Inorganic Perovskite Quantum Dots for Neuromorphic Computing. *Adv. Mater.* **30**, 1802883 (2018).
7. Li, G. *et al.* Photo-induced non-volatile VO₂ phase transition for neuromorphic ultraviolet sensors. *Nat. Commun.* **13**, 1729 (2022).
8. Yang, B. *et al.* Bioinspired Multifunctional Organic Transistors Based on Natural Chlorophyll/Organic Semiconductors. *Adv. Mater.* **32**, 2001227 (2020).
9. Pradhan, B. *et al.* Ultrasensitive and ultrathin phototransistors and photonic synapses using perovskite quantum dots grown from graphene lattice. *Sci. Adv.* **6**,

eaay5225 (2020).

Comment #1:

The gate leakage current of both the transfer characteristics is needed in Figure 2d.

Response:

Thanks for your valuable comments. Following to the reviewer's suggestion, we re-measured the device, and the relevant results are shown in **Figure R1**, which respectively gives the leakage current of the device before and after spin coating of ZnTPP. The blue line represents the leakage current before spin coating of ZnTPP, and the green line represents the leakage current after spin coating of ZnTPP. The leakage current of the device remains relatively stable after spin coating with ZnTPP, indicating its robustness.

Figure R1. Transfer curves of the device before and after spin-coating ZnTPP under a dark condition. (The blue line represents the leakage current before spin coating of ZnTPP, and the green line represents the leakage current after spin coating of ZnTPP.)

Changes in the manuscript:

Corresponding modifications have been made in **Figure 2d**.

Comment #2:

Figure 2f. is missing in the main content, cite wherever necessary.

Response:

Thanks for your valuable comments. We apologize for our oversight in omitting **Figure 2f** in the manuscript. Following to the reviewer's suggestion, we have supplemented **Figure 2f** in the original manuscript and added corresponding references. We further checked the entire text to prevent similar errors from occurring.

Changes in the manuscript:

In line 138-141: This phenomenon is in good agreements with the result of Raman spectroscopy in **Figure 2f**, where the G-mode peak of SWCNTs exhibits a shift of 2 cm^{-1} after spin-coating ZnTPP. In the case of electron donation, the transfer of charge into the SWCNT π system induces a softening effect on the C–C bonds, leading to a redshift in the G peaks.^{25,26}

Comment #3:

Elaborate more on the reason for the blue shift and the decrease in the FWHM (G+) for the heterojunction case.

Response:

Thanks for your valuable comments. Raman spectroscopy offers valuable insights into the impact of doping on SWCNTs. In the study of SWCNTs, the radial breathing mode (RBM) peak is a crucial spectral feature that provides information about the nanotube diameter and phonon structure. The variations in the RBM peak can reveal the influence of molecular interactions on the nanotubes, as these interactions may lead to changes in the electronic properties and structure of the nanotubes. Previous research results indicate that the radial breathing mode (RBM) of carbon nanotubes is highly sensitive to the adsorption of organic molecules on the nanotube surface. The π - π stacking interaction between SWCNTs and phthalocyanine aromatic rings leads to a higher frequency shift of the RBM (Gotovac, S. *et al. Nano Lett.* 7, 583, 2007; Zhang, Y. *et al. J. Am. Chem. Soc.* 127, 17156, 2005). Strong π - π interactions between ZnTPP

and SWCNTs result in a blue shift of the RBM (as observed on the spectra).

The G peak is a characteristic vibration peak of the C-C bonds in SWCNT. Combined with band structure and transfer characteristic analysis, we can conclude that when ZnTPP is spin-coated onto the surface of SWCNTs, it induces electron doping in the nanotubes (n-doping). In the case of electron donors, charge transfer into the SWCNT π system induces a softening effect on the C-C bonds, thereby causing a red shift in the Raman peaks (Mistry, K. S. *et al. ACS Nano* 5, 3714, 2011; Claye, A. S. *et al. Phys. Rev. B* 62, R4845, 2000). This phenomenon unfolds due to the intricate electronic interactions between the dopants and the SWCNTs lattice. Electron-rich dopants donate electrons to the SWCNTs, altering the charge distribution within the π electron cloud. This redistribution of charge affects the vibrational modes of the C-C bonds, leading to observable changes in the Raman spectrum. Specifically, the softening of C-C bonds induced by electron donors causes the Raman peaks to shift towards longer wavelengths, characteristic of a red shift. The observed shifts in the Raman spectrum serve as sensitive indicators of the doping-induced alterations in the SWCNT structure and electronic properties. For example, Smalley *et al.*, used Li, K, and Rb as electron donors, discovering a doping-induced red shift of up to 8 cm^{-1} in the G band due to electron transfer into the SWCNTs (Rao, A. M. *et al. Nature* 388, 257, 1997).

As for the decrease in the FWHM of the G^+ peak, according to previous reports, SWCNTs are easily doped with water oxygen (hole doping) in air, exhibiting characteristics of p-type semiconductors (Li, L.-J. *et al. Nat. Mater.* 4, 481, 2005; Aguirre, C. M. *et al. Adv. Mater.* 21, 3087, 2009), as evident from the transfer characteristic curve as shown in **Figure R2a**. Therefore, the Raman spectrum of SWCNTs directly measured in air can be considered as a result of hole doping. **Figure 2b** shows the FWHM of the G^+ peak of carbon nanotubes at different gate voltages. The different gate voltages can be regarded as different levels of doping (Abdula, D. *et al. Phys. Rev. B* 83, 205419, 2011). Based on the transfer characteristic curve after spin-coating ZnTPP, it is known that ZnTPP transferred some electrons to SWCNTs (weak n-type doping), thereby decreasing the concentration of doped holes. Moreover,

this doping level did not exceed the charge neutrality point, as the transfer characteristic curve did not exhibit significant bipolar and n-type characteristics. According to literature reports, controlling the carrier concentration in semiconductors can reduce the Raman scattering associated with carriers to some extent, thereby narrowing the linewidth of FWHM (Das, A. *et al. Phys. Rev. Lett.* 99, 136803, 2007; Grimm, S. *et al. Carbon* 118, 261, 2017; Abdula, D. *et al. Phys. Rev. B* 83, 205419, 2011).

Figure R2. (a) Transfer curves of the device before and after spin-coating ZnTPP under a dark condition. (b) Dependence of G^+ peak linewidth on doping (varied by gate voltage), citation from the literature (Abdula, D. *et al. Phys. Rev. B* 83, 205419, 2011).

Changes in the manuscript:

In line 140-149: In the case of electron donors, the transfer of charge into the SWCNT π system induces a softening effect on the C–C bonds, leading to a redshift in the Raman peaks. According to previous reports, SWCNTs readily undergo hole doping with water oxygen in air, exhibiting characteristics of p-type semiconductors. The linewidth of FWHM can be controlled by the carrier concentration in SWCNTs. Based on the transfer characteristic curve after spin-coating ZnTPP, it's evident that ZnTPP transfers some electrons to SWCNTs (weak n-type doping), thereby reducing the concentration of hole doping and narrowing the linewidth. Moreover, the radial breathing mode (RBM) exhibited a blue shift of 1.9 cm^{-1} , which was caused by the π - π stacking interaction between SWCNTs and phthalocyanine aromatic rings, as shown in

Figure S5.

Comment #4:

It is recommended to show the transfer characteristics of bare Zn-TPP.

Response:

Thanks for your valuable comments. Following the reviewer's suggestion, we conducted additional experiments (show the characteristics of bare Zn-TPP). Because ZnTPP can be dissolved in other organic solvents such as acetone, we structured the device with a bottom-gate bottom-contact (as shown in **Figure R3a**) configuration to mitigate the impact on ZnTPP during device fabrication. The specific process involved initially preparing the bottom gate electrode on the PI substrate through photolithography, followed by ALD growth of a 50 nm HfO₂ as the dielectric layer. Subsequently, the source and drain electrode was prepared by photolithography process (W=40 μm, L=20 μm). Finally, we coated the ZnTPP solution onto the prepared substrate surface using spin coating. As shown in **Figure R3b, R3c, R3d, R3e**, respectively. To ensure experimental accuracy, we implemented three different spinning conditions: 500 r/min, 1,000 r/min, and 3,000 r/min, each for 30 s. Electrical measurements were conducted on the devices at room temperature. As shown in **Figure R3f**, the devices were unable to conduct under all three conditions, indicating that the ZnTPP thin films prepared by spin coating were inadequate as channel layer materials due to their inability to form homogeneous and dense films. This also demonstrates that ZnTPP primarily functions as a photosensitive layer material in this work.

Figure R3. (a) The diagram of device structure. (b), (c), (d), (e) Optical image of device after ZnTPP spin coating, the spin coating conditions are 500 r/min, 1,000 r/min, 3,000 r/min, respectively. The spinning time was 30 s. (scale bar: 40 μm) (f) Transfer characteristic curves of devices at different spin coating speeds. ($V_{\text{DS}}=1\text{ V}$)

Changes in the manuscript:

In line 165-167: The fabrication of pure ZnTPP FETs was carried out using the spin-coating method. However, the device exhibited poor conductivity, as illustrated in **Figure S7**, further confirming the application of ZnTPP as a photosensitive layer.

Comment #5:

Revise the energy band diagram with the fermi level showing on either side.

Response:

Thanks for your valuable comments. According to previous reports, for nanotubes with diameters larger than 1 nm, the work function is distributed within a narrow range (approximately 0.1 eV), and there is no significant chirality or diameter dependence. This implies that for nanotubes with diameters greater than 1 nm, their work function can be approximated by the work function of graphene, which is approximately 4.66 eV (Shan, B. *et al. Phys. Rev. Lett.* 94, 236602, 2005). **Figure R4a** and **4b** shows a revised version of the energy band diagram, which includes more information including

the fermi level of SWCNT of ~ 4.66 eV and ZnTPP of ~ 3.80 eV (Y. Smets. *et al. J. Chem. Phys.* 139, 044703, 2013).

Figure R4. Energy band gap diagram. (a) Energy band structure of SWCNT and ZnTPP before contact. (b) Energy band diagram under dark condition after contact.

Changes in the manuscript:

Corresponding modifications have been made in **Figure 3g** and **Figure S15** of the manuscript and added corresponding references.

Comment #6:

Schematically illustrate the energy band bending for different stages as discussed in the manuscript.

Response:

Thanks for your valuable comments. Following the reviewer's suggestion, we re-illustrate the energy band bending for different stages as discussed in the manuscript. The band structure distribution of SWCNTs and ZnTPP in the initial state is depicted in **Figure R4a**. At this stage, due to the mismatched band structures of SWCNTs and ZnTPP, and the Fermi level of ZnTPP being higher than that of SWCNTs, ZnTPP transfers some electrons to SWCNTs (Choi, S. *et al. Adv. Mater.* 23, 3979 2011; Wei, S. *et al. Adv. Mater.* 32, 1907527, 2020; Huang, PY. *et al. Nat. Commun* 14, 6736, 2023). Meanwhile, the band of ZnTPP bends upwards, and that of SWCNTs bends downwards, resulting in a typical type I band structure. Upon illumination of the heterojunction, since ZnTPP is the main absorbing layer, numerous hole-electron pairs are rapidly generated in ZnTPP, as shown in **Figure R5a** (corresponding state I). Assisted by the built-in electric field, holes can easily transfer from ZnTPP to SWCNTs,

leading to an increase in channel current. Meanwhile, due to the upward bending of the LUMO band of ZnTPP, which formed a certain barrier, hindering electrons recombination with holes. These electrons continue to be retained in ZnTPP. When the light is removed, these electrons continue to exist, thereby continuously inducing the generation of holes in SWCNTs, as shown in **Figure R5b** (corresponding state II), which is known as the photogating effect, which is the fundamental source of PPC in this work. The direction of electric field generated by the trapped electrons in the ZnTPP is opposite to the built-in field, so the degree of bandgap bending can be reduced.

When applied -2V gate pulse, the density of holes increases instantaneously, allowing the electrons previously blocked to recombine with the holes (Wang, Q. *et al. Sci. Adv.* 4, eaap7916, 2018; Chen, J. *et al. Adv. Mater.* 29, 1702217, 2017), as shown in **Figure R5c** (corresponding state III). Upon pulse removal, the device returns to its initial state, as illustrated in **Figure R5d** (corresponding state IV).

Figure R5. Energy band gap diagram. (a) Energy band diagram under light illumination. (b) The electrons in ZnTPP continuously induce holes generation in SWCNTs. (c) Energy band diagram with -2 V gate pulse. (d) Energy band diagram after the pulse ends.

Changes in the manuscript:

Corresponding modifications have been added to **Figure S15** and **S18** of the manuscript.

Comment #7:

References need on lines 31-33

Response:

Thanks for your valuable comments. Following the reviewer's suggestion, we added relevant references to the corresponding positions in the original manuscript.

Changes in the manuscript:

In line 31-33: However, previous research implemented PPC required additional gate voltages and low temperatures, which need additional energy consumption and cannot maintain stability of optical memory over a wide temperature.^{1,2}

References:

1. Zhang, Z. *et al.* All-in-one two-dimensional retinomorphic hardware device for motion detection and recognition. *Nat. Nanotechnol.* **17**, 27–32 (2022).
2. Hao, J. *et al.* Low-energy room-temperature optical switching in mixed-dimensionality nanoscale perovskite heterojunctions. *Sci. Adv.* **7**, eabf1959 (2021).

Reviewer #2

General comment:

In this manuscript, the authors reported a hybrid ZnTPP/SWCNT heterojunction with strong binding energy. This junction exhibits wide-temperature nonvolatile optical memory and neuromorphic computing with ultra-low power consumption. However, these distinctive features are not fully supported by its related characterizations and data, and the method's uniqueness warrants further investigation. Several questions should be resolved as following.

Response:

We sincerely appreciate the reviewer for taking the time and effort to conduct a thorough review of our paper. Based on the reviewer's professional and constructive comments, new experimental data have been added to the revised manuscript to simultaneously support the conclusion and enhance the readability. With the help of the reviewer, the whole manuscript has been largely improved. In the following, we will address all comments point-to-point and revised the manuscript. We hope that the revised manuscript would address the reviewers' concerns.

Comment #1:

The authors demonstrated that the PPC behavior was attributed to charge trapping between semiconductive SWCNT and ZnTPP. Generally, SWCNTs contain semiconductive and conductive types. What is the purity of semiconductive SWCNTs in this manuscript?

Response:

Thanks for your valuable comments. Following the reviewer's suggestion, we quantified the purity of separated SWCNTs.

In general, the semiconducting purity of SWCNTs solutions can be determined by the absorption spectrum using a reported method (Ding, J. *et al. Nanoscale* 6, 2328, 2014). According to the UV-Vis-NIR results, the absorbance peak ratio ϕ_i can be used

to evaluate the separation purity of semiconducting single-wall carbon nanotubes (s-SWCNTs). The higher the semiconducting purity of s-SWCNTs, the greater the ϕ_i value.

$$\phi_i = A_{\text{CNT}} / (A_{\text{CNT}} + A_{\text{B}})$$

where A_{CNT} represents the enveloping area of the M_{11} and S_{22} bands enclosed by the linear baseline (dotted line), indicating the proportion of metallic single-wall carbon nanotubes (m-SWCNTs) and s-SWCNTs in the sample, as shown in **Figure R6a**. Meanwhile, A_{B} denotes the area covered by the linear baseline in the same region, as shown in **Figure R6b** primarily associated with the presence of amorphous carbon impurities. The purity of s-SWCNTs in our study exceeds 99.99%, which was evidenced by ϕ (0.419) being greater than the value (0.404) reported in a previous study by J. Ding et al.

Figure R6. Characterizing the purity of semiconducting carbon nanotubes through absorption spectrum. (a) The absorption spectra of s-SWCNTs solutions. (b) Absorption spectrum of the purified s-SWCNTs for calculating absorption peak ratio (ϕ).

Furthermore, according to earlier studies, the semiconducting purity of carbon nanotube solutions can also be assessed through the on/off ratio of field-effect transistor (FET) devices (Liu, L. *et al. Science* 368, 850, 2020; Shi, H. *et al. Nat. Electron.* 4, 405, 2021). Therefore, we fabricated highly aligned carbon nanotube films using the previously reported method. The results, as illustrated in **Figure R6a**, demonstrate a

high degree of alignment in the carbon nanotube films based on SEM images, making them suitable for further evaluation of the purity. Subsequently, we utilized TEM to examine the cross-section of the aligned carbon nanotube films, allowing for a statistical assessment of the density of the aligned carbon nanotube films. As depicted in **Figure R6b**, each small circle represents the cross section of a carbon nanotube. The estimated density is approximately 500 tubes / μm for the prepared aligned carbon nanotube films.

Subsequently, leveraging this highly aligned film, we successfully fabricated FET devices, as depicted in **Figure R6c**. SEM images reveal that the channel length of the devices measures 840 nm, with a channel width of 30 μm . A total of 60 sets of FET devices were tested, as shown in **Figure R6d**. Among them, 55 devices exhibited an on/off ratio exceeding 10^3 , while only 5 devices had an on/off ratio below 10^3 . Based on prior research, occurrences of on/off ratios below 10^3 may be resulted from to the presence of metallic carbon nanotubes.

Consequently, the total number of carbon nanotubes tested is calculated to be $500 \text{ (tubes}/\mu\text{m}) \times 60 \times 30 \text{ (}\mu\text{m}) = 90,000 \text{ tubes}$. The semiconductor purity of the carbon nanotubes is estimated to be 99.9994%, deduced from the equation $(1 - 5/90,000) \times 100\%$. Thus, we confidently assert that the purity of the carbon nanotubes employed in our study exceeds 99.9994%.

Figure R7. Characterizing the purity of semiconducting carbon nanotubes through FET device. (a) The SEM images of arrayed carbon nanotube films. Scale bar is 1 μm. (b) TEM image of a CNT array shows a density of 500 CNTs μm^{-1} . Scale bar is 10 nm. (c) SEM images of arrayed carbon nanotube film devices. Scale bar is 4 μm. (d) Transfer curves of 60 original CNT-FETs ($V_{DS} = 1$ V). FETs with 840 nm channel length and 30 μm channel width.

Changes in the manuscript:

In line 128-131: The absorption spectra of the SWCNTs solution are depicted in **Figure S4**. It is evident from the analysis that the semiconductor purity exceeds 99.99%, making it ideal for application as a charge transport layer.

Corresponding modifications have been added to **Figure S4** of the manuscript.

Comment #2:

Due to the advantage of wafer-scale manufacturing process, the reproductivity of each cells is very important, please demonstrate the reproductivity.

Response:

Thanks for your valuable comments. Following the reviewer's suggestion, we conducted electrical measurements on the entire 16-device array across the 4-inch wafer to demonstrate the reproductivity of each cell. The morphology of the devices is illustrated in **Figure R8a**, with the central 16-device array electrodes being intact and suitable for electrical measurements. Prior to the measurements, optical microscopy was employed to inspect the morphology of the devices, as depicted in **Figure R8b**, indicating an excellent uniformity. A total of 1,600 transfer characteristics curves (corresponding to the total device count in the 16 10×10 device arrays) are presented in **Figure R8c**, demonstrating high yield, with each device exhibiting an on/off ratio greater than 10^5 , as shown in **Figure R8d**. The distribution of the on-state currents for 1,600 devices is statistically analyzed, revealing a well-distributed and high on-state current characteristic, as illustrated in **Figure R8e**. Finally, statistical analysis of the threshold voltage for the devices is presented in **Figure R8f**, indicating that most devices have a threshold voltage of around -2 V, emphasizing the uniformity and robustness of the fabricated 4-inch flexible wafer-scale devices.

Figure R8. (a) Optical photograph of a 4-inch flexible wafer arrays peeled off from the silicon substrate. (b) Optical photograph of partial device from the array. (c) The transfer characteristic curves of 1,600 SWCNTs/ZnTPP phototransistors were measured under dark, room temperature conditions. These devices exhibited uniform electrical performance at $V_{\text{DS}} = 1$ V. (d) The statistical distribution of the on/off ratio. (e) The statistical distribution of on-state current. The red curve is an normal curve. (f) The statistical distribution of threshold voltage (V_{th}). The red curve is an normal curve.

Changes in the manuscript:

In line 307-310: To validate the uniformity and stability of the device array, we

subjected 1,600 transistors in the array to transfer characteristic curve tests, as depicted in **Figure S20**. We recorded the corresponding on/off ratio, on-state current, and threshold voltage, revealing an excellent uniformity across our devices.

Corresponding modifications have been added to **Figure S20** of the manuscript.

Comment #3:

In the figure 3f, the photoresponse properties are highly dependent on the working temperature, thus the title with wide-temperature optical memory is seem to be inaccurate. Meanwhile, the recyclability of temperature-dependent photoresponse should be measured.

Response:

Thanks for your valuable comments. Following the reviewer's valuable suggestion, we have adjusted the title by removing "wide-temperature." The title is modified to be: *"Ultra-Low Power Carbon Nanotube/Porphyrin Synaptic Arrays for Optical Memory and Neuromorphic Computing"* .

Additionally, based on the reviewer's suggestion, we conducted measurements on the repeatability of light response at different temperatures. As shown in **Figure R7**, the repeatability of light response was measured separately at temperatures ranging from 77 K to 400 K, because the device exhibits an excellent optical memory capability within this temperature range, gate pulse voltage assistance is required to eliminate the memory effect, during testing, a gate pulse voltage of -2V was applied to remove the memory effect of the device. After the device was stabilized for 10 s, the same light pulse was irradiated once again, followed by the continuation of gate pulse voltage application after another 10 seconds, in this cyclic manner. Five rounds of light response measurements were carried out at different temperatures, all exhibiting a good repeatability.

Figure R9. The repeatability of light response at different temperatures. Optical pulses and electrical pulses were used for writing and erasing, respectively. Both light pulse width and electrical pulses width are 0.2 s. ($V_{DS} = 0.1$ V, $P = 1$ mW/cm²)

Changes in the manuscript:

New title: *Ultra-Low Power Carbon Nanotube/Porphyrin Synaptic Arrays for Optical Memory and Neuromorphic Computing*

In line 263-265: The repeatability of light response was measured separately at temperatures ranging from 77 K to 400 K, as shown in **Figure S17**, all exhibiting good repeatability.

Corresponding modifications have been added to **Figure S17** of the manuscript.

Comment #4:

The authors employ SEM image to evaluate the uniform of film, and XRD or Raman spectra from different areas should be also provided.

Response:

Thanks for your valuable comments. Following the reviewer's suggestion, we divided the 4-inch wafer into four sections (before spin-coating ZnTPP), as shown in the **Figure R10a**. 100 random positions were selected for Raman measurements. The major characteristic peaks of SWCNTs, including the RBM peak near 171 cm⁻¹, the

D peak near 1350 cm^{-1} , and the G^+ peak near 1594 cm^{-1} (with single-walled tubes typically exhibiting a split G peak into G^+ and G^- peaks), are clearly observed. The detailed characteristics of the RBM peak are depicted in **Figure R10a**, where the RBM peak positions of the tested 100 points are approximately $171\pm 0.6\text{ cm}^{-1}$. Similarly, the detailed features of the G^+ peak are shown in **Figure R10b**, with the RBM peak positions of the tested 100 points being approximately $1594\pm 0.4\text{ cm}^{-1}$. These results indicate an excellent uniformity of the prepared wafer-scale SWCNTs film, making it suitable for the fabrication of wafer-scale devices integrated. Additionally, we have provided additional SEM images depicting a broader range of locations, as shown in the **Figure R11**. We selected nine distinct positions for SEM characterization, and the results demonstrate a remarkably uniform morphology at different sites of the film. A sufficiently high homogeneity in the morphology among these sites further corroborate the uniformity of the films in the 4-inch devices that we prepared.

Figure R10. Raman Characterization of 4-inch SWCNT film. (a) The RBM region of carbon nanotube films at 100 different positions. (b) The G region of carbon nanotube films at 100 different positions.

Figure R11. SEM Characterization of carbon nanotube films at 9 different positions.

Changes in the manuscript:

In line 310-313: 100 random positions were selected for Raman measurements to evaluate the uniform of film, and the results are illustrated in **Figure S21**, where the RBM peak positions of the tested 100 sites are approximately $171\pm 0.6\text{ cm}^{-1}$. Similarly, the detailed features of the G^+ peak are approximately $1594\pm 0.4\text{ cm}^{-1}$. SEM characterizes more regions at the same time in **Figure S22**, showing a large-area uniformity of the 4-inch wafer.

Corresponding modifications have been added to **Figure S21** and **S22** of the manuscript.

Comment #5:

To highlight the nonvolatile feature of the heterojunction device, the authors are suggested to prolong the retention time.

Response:

Thanks for your valuable comments. Following the reviewer's suggestion, we re-measured the retention time of the device, as shown in **Figure R12** extending it to 2×10^4 s. The device consistently maintained highly stable storage characteristics. Even after being powered off for 10 h and remeasured, only a small amount of current loss was observed, thus fully demonstrating the non-volatile storage properties of the device. We also compared our results with recent literature findings in terms of retention time, as shown in **Table R2**, the retention time of SWCNTs/ZnTPP phototransistor is the longest.

Figure R12. Optical pulse intrigues a PPC state. ($V_{DS}=1V$, $P= 1 \text{ mW/cm}^2$)

Table R2. Comparison of devices with persistent photoconductivity for latest Literature Studies. (RT: Room temperature)

Active materials	Substrate	Temperature (K)	Time (s)	Energy Per Spike (fJ)	Scale	Reference
SWCNT/ FAPbBr ₃	Si/SiO ₂	~50-RT	$\sim 5 \times 10^3$	7.4	Single	Ref.1
BP/CdS	Si/SiO ₂	RT	$> 5 \times 10^3$	4.8	Single	Ref.2
MoS ₂ /PbS	Si/SiO ₂	<200	$> 1 \times 10^4$	138000	Single	Ref.3
IGZO/PV K	PET/Al ₂ O ₃	RT	$> 1 \times 10^4$	Not mentioned	~ 2 cm \times 2 cm	Ref.4
MoS ₂ /PO _x	Si/SiO ₂	80-300	$> 1 \times 10^4$	Not mentioned	Single	Ref.5
Pentacene / CsPbBr ₃	Si/SiO ₂	RT	$\sim 3 \times 10^3$	~ 140	Single	Ref.6

VO ₂	Si/SiO ₂	RT	~4×10 ³	Not mentioned	2-inch	Ref.7
PDPP4T/ chlorophy ll	Si/SiO ₂	RT	~4×10 ²	0.25	Single	Ref.8
Graphene/ PQD	Si/SiO ₂	RT	~3×10 ³	37000	Single	Ref.9
SWCNT/ ZnTPP	PI/HfO₂	77-400	>2×10⁴	0.065	4-inch	This work

Changes in the manuscript:

In line 36-37: Remarkably, it demonstrated nonvolatile storage for up to 2×10^4 s, without additional gate voltage.

Corresponding modifications have been added to **Figure 3c** of the manuscript.

References:

1. Hao, J. *et al.* Low-energy room-temperature optical switching in mixed-dimensionality nanoscale perovskite heterojunctions. *Sci. Adv.* **7**, eabf1959 (2021).
2. Zhu, C. *et al.* Optical synaptic devices with ultra-low power consumption for neuromorphic computing. *Light Sci Appl* **11**, 337 (2022).
3. Wang, Q. *et al.* Nonvolatile infrared memory in MoS₂/PbS van der Waals heterostructures. *Sci. Adv.* **4**, eaap7916 (2018).
4. Wei, S. *et al.* Flexible Quasi-2D Perovskite/IGZO Phototransistors for Ultrasensitive and Broadband Photodetection. *Adv. Mater* **32**, 1907527 (2020).
5. Liu, C. *et al.* Realizing the Switching of Optoelectronic Memory and Ultrafast Detector in Functionalized-Black Phosphorus/MoS₂ Heterojunction. *Laser & Photonics Rev* **17**, 2200486 (2023).
6. Wang, Y. *et al.* Photonic Synapses Based on Inorganic Perovskite Quantum Dots for Neuromorphic Computing. *Adv. Mater.* **30**, 1802883 (2018).
7. Li, G. *et al.* Photo-induced non-volatile VO₂ phase transition for neuromorphic ultraviolet sensors. *Nat. Commun* **13**, 1729 (2022).
8. Yang, B. *et al.* Bioinspired Multifunctional Organic Transistors Based on Natural Chlorophyll/Organic Semiconductors. *Adv. Mater.* **32**, 2001227 (2020).

9. Pradhan, B. *et al.* Ultrasensitive and ultrathin phototransistors and photonic synapses using perovskite quantum dots grown from graphene lattice. *Sci. Adv.* **6**, eaay5225 (2020).

Comment #6:

The mechanism of nonvolatile behavior should be explained in more details.

Response:

Thanks for your valuable comments. The band structure distribution of SWCNTs and ZnTPP in the initial state is depicted in **Figure R13a**. At this stage, due to the mismatched band structures of SWCNTs and ZnTPP, and the Fermi level of ZnTPP being higher than that of SWCNTs, ZnTPP transfers some electrons to SWCNTs (Choi, S. *et al. Adv. Mater.* 23, 3979 2011; Wei, S. *et al. Adv. Mater.* 32, 1907527, 2020; Huang, PY. *et al. Nat. Commun* 14, 6736, 2023). Meanwhile, the band of ZnTPP bends upwards, and that of SWCNTs bends downwards, resulting in a typical type I band structure, as shown in **Figure R13b**. Upon illumination of the heterojunction, since ZnTPP is the main absorbing layer, numerous hole-electron pairs are rapidly generated in ZnTPP, as shown in **Figure R13c** and **Figure R14b**. Assisted by the built-in electric field, holes can easily transfer from ZnTPP to SWCNTs, leading to an increase in channel current. Meanwhile, due to the upward bending of the LUMO band of ZnTPP, which formed a certain barrier, hindering electrons recombination with holes. These electrons continue to be retained in ZnTPP. When the light is removed, these electrons continue to exist, thereby continuously inducing the generation of holes in SWCNTs, as shown in **Figure R13d** and **Figure R14c**, which is known as the photogating effect, which is the fundamental source of PPC in this work.

Figure R13. Energy band gap diagram. (a) Band structure of SWCNT and ZnTPP (b) Energy band diagram under dark. (c) Energy band diagram under light illumination. (d) The electrons in ZnTPP continuously induce holes generation in SWCNTs.

Figure R14. (a), (b), (c) The charge-trapping state of the device in dark condition, under light illumination and after the cessation of illumination, respectively. (hole: white voids, electron: red voids)

Changes in the manuscript:

Corresponding modifications have been added to **Figure S16 and S18** of the manuscript.

Comment #7:

In the figure 4h, the comparison of the device power consumptions with latest reported literatures should summarize the results of self-powered works.

Response:

Thanks for your valuable comments. Following the reviewer's suggestion, we have incorporated the results of self-powered works into the comparative data, as shown in **Figure R15**. It is worth noting that self-powered devices are typically classified into two categories: gate self-powered, such as triboelectric nanogenerator (TENG), and self-biased devices. For gate self-powered devices, significant additional energy consumption is required, making it challenging to estimate the actual power consumption (Yu, J. *et al. Sci. Adv.* 7, eabd9117, 2021; Liu, Y. *et al. Nat. Commun* 13, 7917, 2022). For self-biased devices, sometimes excessive energy is needed to initiate them (Kumar, M. *et al. Nano Energy* 89, 106471, 2021). In other cases, additional energy is not required to trigger the device, i.e., $V_{DS}=0$ V. Therefore, according to mathematical formulas: $E = V_{DS} \times I_{EPSC} \times t$. Their theoretical power consumption is zero (Kumar, M. *et al. Adv. Mater.* 31, 1903095, 2019). To avoid ambiguity, we have made certain modifications to the relevant statements in the original manuscript.

Figure R15. The comparison of the device power consumptions with latest reported works.

Changes in the manuscript:

In line 299-303: Compared to those investigated in **non-self-powered** works in the past, as shown in **Figure 4h**, our photonic synaptic device exhibits the lowest power consumption and is significantly lower than the power consumption per synapse in human brain activity (1-100 fJ), indicating the device's enormous potential in

developing ultra-low-power neuromorphic chips.

Comment #8:

The optical absorption spectra of ZnTPP is at 422 nm and 550 nm, while the authors choose 395 nm as the pulse light, why?

Response:

Thanks for your valuable comments. According to the absorption spectrum and band structure of ZnTPP, it is evident that ZnTPP absorbs throughout the entire visible light spectrum. As mentioned by the reviewer, it exhibits prominent absorption peaks in the Soret band near 422 nm and the Q band near 550 nm. According to previous reports. (Winkelmann, C. B. *et al. Nano Lett.* 7, 1454, 2007), the closer the wavelength of light is to 422 nm, the greater the photocurrent generated by excitation. However, since our light source is limited to only few wavelengths, including 395 nm, 450 nm, 532 and 620 nm, we employed various sources to stimulate photocurrent in the devices. The results, as depicted in the **Figure R16**, demonstrate that the SWCNTs/ZnTPP phototransistors exhibit good response across the entire visible wavelength region. Notably, the 395 nm light source yielded the largest response, with the greatest photocurrent generated under identical pulse duration and power conditions. Consequently, in this study, we selected 395 nm light as the pulse light source. This selection does not compromise the outcome or the underlying mechanisms of the experiment.

Figure R16. The photocurrent generated by the device under different wavelength illumination conditions. ($P_{\text{light}}=0.5 \text{ mW/cm}^2$, $V_{\text{DS}} = 1\text{V}$)

Changes in the manuscript:

In line 161-163: Four different wavelengths of light are used to irradiate phototransistor, and it is found that the photocurrent generated under 395 nm irradiation is the largest, as shown in **Figure S6**, so 395 nm is chosen as the laser light source.

Corresponding modifications have been added to **Figure S6** of the manuscript.

Comment #9:

In the figure 5b, the authors demonstrate that the data is obtained by encoding RGB array from pixel values, thus the EPSC measurement at different wavelength of pulse light should be provided.

Response:

Thanks for your valuable comments. Following the reviewer's suggestion, we conducted measurements of EPSC using different light sources, R (620 nm), G (532 nm), and B (450 nm), as depicted in **Figure R17a** and **R17b**. The pulse width used was 0.2s, and the light sources were of the same power. Additionally, we separately evaluated the non-volatile storage characteristics of the devices excited by these three wavelength light sources, as illustrated in **Figure R17c**. Within the tested 2×10^3 s, it was evident that all three laser wavelengths utilized could achieve PPC, demonstrating

the versatility of our devices in the visible wavelength region. Furthermore, in **Figure R17d**, the pulse-switching characteristics of optical potentiation and electrical depression were investigated in the SWCNT/ZnTPP phototransistor under different light wavelength. The channel current of the transistor can be reversibly switched between high and low-current states under different light wavelength, demonstrating a high stability.

Figure R17. (a) Schematic diagram of the SWCNTs and ZnTPP heterojunction device. (b) The photocurrent generated by the device under different wavelength illumination conditions. ($P_{\text{light}}=0.5 \text{ mW/cm}^2$, $V_{\text{DS}} = 1\text{V}$), (c) Optical pulse intrigues a PPC state under 450 nm laser, 532 nm laser, 620 nm laser, respectively. ($P_{\text{light}}=0.5 \text{ mW/cm}^2$ and $V_{\text{DS}} = 1\text{V}$) (d) Light-controlled LTP (light intensity 0.5 mW/cm^2 , duration of 1 s, spaced 2 s apart) and V_{G} -controlled LTD ($V_{\text{G}}=-1.5\text{V}$, duration of 1 s, spaced 1s apart) for 50 pulses under 450 nm laser, 532 nm laser, 620 nm laser, respectively. $V_{\text{DS}} = 1 \text{ V}$.

Changes in the manuscript:

In line 180-186: 450 nm, 532 nm and 620 nm lasers were also used as the light sources to determine the PPC characteristics of the phototransistor, as shown in **Figure S9a**, **S9b** and **S9c** which remained stable over the course of 2000 s, highlighting the universality of the device in the visible light range. Furthermore, in **Figure S9d**, the pulse-switching characteristics of optical potentiation and electrical depression were investigated in the SWCNTs/ZnTPP phototransistor under different light wavelength. The channel current of the transistor can be reversibly switched between high and low-current states under different light wavelength, demonstrating a high stability

Corresponding modifications have been added to **Figure S9** of the manuscript.

REVIEWER COMMENTS

Reviewer #1 (Remarks to the Author):

The author has thoroughly addressed all the points mentioned in the previous review process and incorporated them into the revised manuscript. Therefore, I hope this paper will be accepted in its current format.

Reviewer #2 (Remarks to the Author):

In the revised manuscript, the authors deleted the demonstration of wide temperature in the title. One of the motivation of this work is to address the high stability of optical memory over a wide temperature in the abstract. However, to my opinion, this issue cannot be addressed according to the Figure 3f. Based on this results, the temperature have large negative influence on optical memory performance, this trend normally presents in the organic or organic-inorganic hybrid system. Another issue is that the high-purity dispersion of SWCNTs was obtained with the assistance of polymeric PCz. According to the illustration of device fabrication in the experiment, there is no treatment to remove the PCz, meaning PCz still residual in the SWCNT network. How about the effect of PCz on the performance? The structure of device and related mechanism should be modified.

Response Letter to Reviewers' Comments

We sincerely appreciate the reviewers for carefully reading our manuscript '*Ultra-Low Power Carbon Nanotube/Porphyrin Synaptic Arrays for Optical Memory and Neuromorphic Computing*' (NCOMMS-24-04786A) and providing constructive comments. According to the reviewers' comments, we have carefully revised our manuscript and provided more detailed data to improve the quality and readability of the manuscript. We have reorganized the manuscript and supplementary materials, revised relevant descriptions, and added new data. With the help of the reviewers, we believe that the submitted version has been significantly improved. Below are the point-by-point responses to each comment.

The corresponding revisions concerning comments have been provided and highlighted in red in the revised manuscript and supplementary information. The corresponding responses of the reviewers are marked by blue words.

Reviewer #1

General comment:

The author has thoroughly addressed all the points mentioned in the previous review process and incorporated them into the revised manuscript. Therefore, I hope this paper will be accepted in its current format.

Response:

We are very glad to see that you are satisfied with our revision. We sincerely thank you for the valuable time you have spent reviewing our manuscript and providing insightful comments to help significantly improve the quality of our work.

Reviewer #2

General comment:

In the revised manuscript, the authors deleted the demonstration of wide temperature in the title. One of the motivation of this work is to address the high stability of optical memory over a wide temperature in the abstract. However, to my opinion, this issue cannot be addressed according to the Figure 3f. Based on this results, the temperature have large negative influence on optical memory performance, this trend normally presents in the organic or organic-inorganic hybrid system. Another issue is that the high-purity dispersion of SWCNTs was obtained with the assistance of polymeric PCz. According to the illustration of device fabrication in the experiment, there is no treatment to remove the PCz, meaning PCz still residual in the SWCNT network. How about the effect of PCz on the performance? The structure of device and related mechanism should be modified.

Response:

We appreciate the valuable time the reviewer took to review our manuscript and provide constructive feedback. We wholeheartedly agree with the reviewer's assessment that our description of the device's capacity for high stability optical storage across a broad temperature spectrum lacks precision.

As noted by the reviewer, our material system belongs to the organic-inorganic hybrid system. Temperature significantly affects the storage characteristics, with higher temperatures leading to lower retention ratio of photocurrent. Our intended meaning was that the device exhibits a certain storage effect within the range of 77 K to 400 K, rather than saying that between 77 K to 400 K have the same storage effect.

The device shows varying storage performance at varying temperatures. It's worth noting that the current of SWCNTs/ZnTPP phototransistor increase with increasing temperature, which may stem from the additional current contribution of thermally excited-electrons. This is a key factor contributing to the variation in photocurrent of the device following illumination at different temperatures. This phenomenon is likely

attributable to some localized electrons surmounting potential barriers due to strong thermal fluctuations at elevated temperatures, consequently weakening or even nullifying the photogating effect, a phenomenon observed in previous report. (*Sci. Adv.* 4, eaap7916, 2018)

Additionally, the previous representation of **Figure 3f** was not sufficiently clear and intuitive. We have revised it for better clarity and comprehensibility by applying logarithmic transformation to all data points. Simultaneously, we conducted tests on the device's photocurrent retention characteristics at elevated temperatures, as illustrated in **Figure R2**. We observed that the device exhibited discernible photocurrent retention even as the temperature reached 400 K. However, as the temperature further increased to 450 K or 500 K, the amplified thermal disturbance resulted in the device essentially losing its ability to retain the photocurrent. The results demonstrate that the device exhibited persistent photoconductivity (PPC) capability between 77 K to 400 K, which contrasts with the previously reported nonvolatile phototransistors that only operated at low temperatures or room temperature, as shown in **Table S1**.

The device itself is more characterized by synaptic plasticity and maintains certain persistent photoconductive properties at different temperatures. More importantly, to address the reviewers' concerns, we have also changed the title to "Ultra-Low Power Carbon Nanotube/Porphyrin Synaptic Arrays for Persistent Photoconductivity and Neuromorphic Computing"

Figure R1. PPC phenomena of the device induced by the same light pulse at varying

temperatures.

Figure R2. PPC phenomena of the device induced by the same light pulse at varying temperatures.

To address your concern regarding the "high stability of optical memory over a wide temperature" issue, we have revised the corresponding statements in the Abstract, Introduction, and Conclusions sections.

Changes in the manuscript:

~~In line 27-29: Developing devices with a wide-temperature range optical memory stability and ultra-low power consumption remains a significant challenge for optical synaptic devices used in neuromorphic computing.~~

Developing devices with a wide-temperature range **persistent photoconductivity (PPC)** and ultra-low power consumption remains a significant challenge for optical synaptic devices used in neuromorphic computing.

~~In line 32-33: However, previous research implemented PPC required additional gate voltages and low temperatures, which need additional energy consumption and cannot maintain a high stability of optical memory over a wide temperature.~~

However, previous research implemented PPC required additional gate voltages and low temperatures, **which need additional energy consumption and PPC cannot be achieved across a wide temperature range.**

~~In line 41-42: This tunable and stable wide-temperature storage capability holds promise for ultra-low power neuromorphic computing and optical storage applications.~~

This tunable and stable wide-temperature PPC capability holds promise for ultra-low-power neuromorphic computing.

~~In line 77-78: Therefore, the device can reliably store optical signals over a wide temperature range (77 K-400 K) and achieve multi-state storage without the need for additional gate voltages.~~

Therefore, the device can achieve PPC over a wide temperature range (77 K-400 K) without additional energy consumption.

~~In line 203-204: The results demonstrate that the device exhibited a high charge storage capability under different temperatures.~~

The results demonstrate that the device can achieve PPC under different temperatures.

~~In line 306-309: Table S1 provides the detailed results of device tests here and in previous works, demonstrating a significant advantage of SWCNTs/ZnTPP over other material systems in terms of a wide temperature optical memory and low power consumption.~~

Table S1 provides detailed results of device testing in this study and previous work, demonstrating the superior capability of SWCNTs/ZnTPP in achieving PPC at wide temperature ranges and low power consumption compared to other material systems.

~~In line 401-402: Within a wide temperature ranging between 77 K and 400 K, the device demonstrates a durable light memory.~~

The device can achieve persistent photoconductivity between 77 K to 400 K.

Corresponding modifications have been added to **Figure 3f** of the manuscript.

Table R1. Comparison of devices with persistent photoconductivity for latest Studies. (RT: Room temperature)

Active materials	Substrate	Temperature (K)	Time (s)	Reference
SWCNT/FAPbBr ₃	Si/SiO ₂	~50-RT	~5×10 ³	Ref.1
BP/CdS	Si/SiO ₂	RT	>5×10 ³	Ref.2
MoS ₂ /PbS	Si/SiO ₂	<200	>1×10 ⁴	Ref.3
MoS ₂ /PO _x	Si/SiO ₂	80-300	>1×10 ⁴	Ref.4
Pentacene/CsPbBr ₃	Si/SiO ₂	RT	~3×10 ³	Ref.5
PDPP4T/chlorophyll	Si/SiO ₂	RT	~4×10 ²	Ref.6
Graphene/PQD	Si/SiO ₂	RT	~3×10 ³	Ref.7
SWCNT/ZnTPP	PI/HfO ₂	77-400	>2×10 ⁴	This work

References:

1. Hao, J. *et al.* Low-energy room-temperature optical switching in mixed-dimensionality nanoscale perovskite heterojunctions. *Sci. Adv.* **7**, eabf1959 (2021).
2. Zhu, C. *et al.* Optical synaptic devices with ultra-low power consumption for neuromorphic computing. *Light. Sci. Appl.* **11**, 337 (2022).
3. Wang, Q. *et al.* Nonvolatile infrared memory in MoS₂/PbS van der Waals heterostructures. *Sci. Adv.* **4**, eaap7916 (2018).
4. Liu, C. *et al.* Realizing the Switching of Optoelectronic Memory and Ultrafast Detector in Functionalized-Black Phosphorus/MoS₂ Heterojunction. *Laser & Photonics Rev* **17**, 2200486 (2023).
5. Wang, Y. *et al.* Photonic Synapses Based on Inorganic Perovskite Quantum Dots for Neuromorphic Computing. *Adv. Mater.* **30**, 1802883 (2018).
6. Yang, B. *et al.* Bioinspired Multifunctional Organic Transistors Based on Natural Chlorophyll/Organic Semiconductors. *Adv. Mater.* **32**, 2001227 (2020).
7. Pradhan, B. *et al.* Ultrasensitive and ultrathin phototransistors and photonic synapses using perovskite quantum dots grown from graphene lattice. *Sci. Adv.* **6**, eaay5225 (2020).

As for PCz, the presence of the polymer PCz on the surface of SWCNTs can indeed

significantly influence the performance of devices, constituting a critical concern. In the original manuscript, a method for the removal of PCz has been mentioned (**in Lines 430-431, NCOMMS-24-04786A**). Below is a screenshot of the section on polymer removal from the original manuscript. The polymer means PCz.

429 method. With 2 h of deposition, a high-density and random network of SWCNTs can
430 be obtained. The polymer on the surface of SWCNTs was then removed by the
431 tetrahydrofuran cleaning. The clean SWCNTs network film was then patterned using
432 the standard photolithography techniques, with excessive SWCNTs removed by O₂
433 plasma etching. Source and drain electrodes (Ti/Au, 10/50 nm) were fabricated based

However, due to our oversight, we inadvertently omitted the corresponding data in the manuscript, leading to a misunderstanding by the reviewer. We sincerely apologize for this oversight. In the latest revised version, we have included the relevant data to confirm the effective removal of PCz from the surface of SWCNTs.

In the original manuscript, we employed a highly effective solution-based method for PCz removal, as previously reported by our group (*ACS Appl. Mater. Interfaces* 9, 15719, 2017). In our prior study, we demonstrated the excellent solubility of PCz in tetrahydrofuran (THF) solution. Consequently, we immersed the prepared films in THF solution and washed them three times to effectively remove residual PCz. **Figure R3a** illustrates the chemical structure of PCz. Initially, we conducted X-ray photoelectron spectroscopy (XPS) characterization of the films before and after THF washed. As depicted in **Figure R3a**, noticeable nitrogen elements were detected on the surface of films without THF washed, whereas no nitrogen elements were detected on THF-washed film surfaces, indicating the effective removal of PCz from SWCNTs surfaces during washing. Raman spectroscopy results further corroborated this conclusion. We measured pure PCz, films of SWCNTs without THF washed, and films after THF washed, as shown in **Figure R3b**. Pure PCz exhibited characteristic peaks around 1624 cm⁻¹. Typical PCz peaks were observed on films without THF washed, while no PCz characteristic peaks were detected on THF-washed film surfaces, confirming the efficacy of THF washed in polymer removal. Additionally, examination of 100 sets of

Raman results from **Figure S22** revealed no significant PCz signals, indicating uniform and thorough cleaning of the entire 4-inch wafer film.

Furthermore, using the same process, we fabricated transistor devices by using films with and without THF washed, testing a total of 20 devices. As depicted in **Figure R3c**, the results indicated significantly lower on-state currents for devices without washed compared to those with washed, suggesting that the presence of PCz severely affects the direct contact between metal electrodes and SWCNTs, resulting in increased contact resistance and reduced on-state current. Devices subjected to THF washed exhibited a more uniform distribution of on-state currents compared to those unwashed, facilitating wafer manufacturing. It's important to note that this differs from the transfer characteristics curve provided in the supplementary information, as the testing here focused on pure SWCNTs, while the supplementary information presented SWCNT/ZnTPP transfer characteristics curves.

Subsequently, we spin-coated identical amounts of ZnTPP on device surfaces, evaluating their retention characteristics, as shown in **Figure R3d**. When subjected to the same pulsed light on both washed and unwashed devices, it was observed that the washed devices maintained excellent retention characteristics within the same test duration, whereas the unwashed devices exhibited almost no retention characteristics, with photocurrent decaying to the initial value within a short time. This indicates that the presence of PCz severely hinders direct contact between ZnTPP and SWCNT, making it difficult to form high-quality interfaces, thus indirectly demonstrating THF's excellent cleaning effect on PCz. The entire experimental outcome is predicated on the effective removal of PCz.

Figure R3. (a) The XPS results of surface N element on SWCNT thin films before and after THF washed. (b) The Raman results of pure PCz, and SWCNT thin films before and after THF washed. (c) The transfer characteristic curves of 10 SWCNT thin-film transistors with THF-washed and unwashed 10 SWCNT thin-film transistors unwashed were measured under dark, room temperature conditions. (d) The retention characteristics of the transistor with THF-washed and that unwashed were measured separately. ($P_{light}=0.5 \text{ mW/cm}^2$, $V_{DS} = 1V$.)

Supplementary Figure S22. Raman Characterization of 4-inch SWCNT film. (a) The

RBM region of SWCNT films at 100 different positions. (b)The G region of SWCNT films at 100 different positions.

Changes in the manuscript:

In line 180-183: We also evaluated the influence of PCz on the PPC characteristics of the device before and after washed. As shown in **Figure S9**, indicate that the PCz significantly affects the device's on-state current and PPC characteristics. Therefore, PCz cleaning before device fabrication is crucial.

In Supplementary Figure S1: (b) Atomic layer deposition (ALD) of a 70 nm HfO₂ layer at 250°C, followed by interconnect window opening using ion beam etching (IBE) technique, and subsequent deposition of a thin film of SWCNTs by solution-based methods, the PCz on the surface of SWCNTs film was cleaned with tetrahydrofuran;

Corresponding modifications have been added to **Figure S9** of the manuscript.

We hope these answer your questions. We sincerely thank you again for the valuable time you have spent reviewing our manuscript and providing insightful comments to help significantly improve the quality of our work.

REVIEWERS' COMMENTS

Reviewer #2 (Remarks to the Author):

Accept in its current state.

Response Letter to Reviewers' Comments

We sincerely appreciate the reviewers for carefully reading our manuscript '*Ultra-Low Power Carbon Nanotube/Porphyrin Synaptic Arrays for Persistent Photoconductivity and Neuromorphic Computing*' (NCOMMS-24-04786B) and providing constructive comments.

Reviewer #2

General comment:

Accept in its current state.

Response:

We are very glad to see that you are satisfied with our revision. We sincerely thank you for the valuable time you have spent reviewing our manuscript and providing insightful comments to help significantly improve the quality of our work.